# A 240-Year History of Avalanche Risk in the Vosges Mountains Based on Nonconventional Sources

Florie Giacona[1], Nicolas Eckert[2], Brice Martin[1]

[1] Centre de recherche sur les Économies les Sociétés les Arts et les Techniques, Université de Haute Alsace, Mulhouse, 68093, France
[2] UR ETNA, Irstea Grenoble / Université Grenoble Alpes, Saint Martin d'Hères, 38402, France

*Correspondence to*: Florie Giacona (florie.giacona@uha.fr)

**Abstract.** Despite the strong societal impact of mountain risks, their systematic documentation remains poor. Therefore, snow avalanche chronologies exceeding several decades are exceptional, especially in medium-high mountain ranges. This article implements a combination of historical and geographical methods leading to the reconstruction, at the scale of the entire Vosges Mountains (north-east of France), of more than 700 avalanches that have occurred since the late eighteenth century on 128 paths. The clearly episodic nature of the derived geo-chronology can be explained by three interrelated factors that have changed together over time: the body and reliability of sources, social practices conditioning the vulnerability, and the natural hazard itself. Finally, the geo-chronology reflects the changes in the meaning of the hazard in social space. Specifically, the event which could be retrieved from the historical sources is an aspect of the interaction between society and its environment. These results confirm the role of the historian in contextualising and evaluating such data. It transforms these data into information that is relevant for mitigating risk and understanding its change over time. The work also demonstrates the usefulness of constructing an original database from a diverse suite of historical data and field investigations. This approach could be applied to other risk phenomena in the frequent situation where archival data are sparse.

## 1 Introduction

Databases summing up observations of past events play a crucial role in evaluating the expected frequency and severity of natural hazards. However, they are typically sparse in their coverage. Their comprehensiveness is hampered by multiple factors, including: i) a close link to vulnerability (especially events with human casualties), which results in an under-representation of geophysically significant events that did not cause damage or fatalities; ii) a relatively short temporal coverage; and iii) incomplete spatial coverage that excludes large regions. Such factors are particularly prevalent in developing countries where there is often an absence of relevant records (e.g. von Kotze and Holloway, 1996). In view of the considerable socio-economic impact of hydro-climatic phenomena, these limitations of existing records are paradoxical (IPCC, 2012).

Avalanche risk is a characteristic illustration of this situation. Snow avalanches strongly impact permanent and temporary populations of mountain areas in winter (*e.g.* McClung and Schearer, 2006; Schweizer *et al.*, 2003), causing death and destruction (buildings, tourism infrastructure, power lines, forest stands), road cut-offs, and, more widely, losses related to an indirect vulnerability of mountain societies (the negative impact of such destructive events on tourism, etc.). Databases concerning the hazard and/or the risk have been developed in many countries, but they are heterogeneous in terms of information content (Laternser and Schneebeli, 2002; Borrel and Brunet, 2006; Bourova *et al.*, 2016). In addition, avalanche chronologies exceeding a few decades remain exceptional (Corona *et al.*, 2013; Schläppy *et al.*, 2014), and those not biased towards the events causing damage, or those triggered artificially even more so (Schneebeli *et al.*, 1997; Eckert *et al.*, 2013; Podolskiy *et al.*, 2014). Avalanche chronologies are even totally lacking in some mountain areas, especially in medium-high mountain ranges that are often neglected in favour of high mountains. This is the situation for the Vosges Mountains, a medium-high range located in north-eastern France, whose significance in terms of avalanche risk is overshadowed by the Alps. Indeed, the latter region has undergone systematic observation of avalanches since the late nineteenth century and has been the focus of substantial interest regarding snow avalanche science and risk engineering (Mougin, 1922; Giacona *et al.*, in press).

Yet, researchers and risk managers need long data series to detect changes in risk with time (due to both social and environmental factors) and estimate the relevant probability of event occurrence. That is, long data series allow one to understand the dynamics of the phenomenon (return period), to define reference scenarios and to contextualise them. Historical analysis is now widely used to reconstruct chronologies and specify the characteristics of past events, particularly in terms of flood risk (Stedinger and Cohn, 1986). However, avalanche risk stands out as an exception to this pattern, with very few studies on this topic by historians: for example, Granet-Abisset and Brugnot (2002) in France, Bruno (2013) in the former Soviet Union, Laely (1984) in Switzerland and di Stefano (2013) in the eastern United States and Canada. In addition, these rare studies have frequently been conducted within a framework that combines snow avalanches with other types of gravitational hazards, and at the spatial scale of a village or a mountain region (Leone, 2006; Granet-Abisset and Montredon, 2007; Favier and Granet-Abisset, 2000; Favier and Remacle, 2007). However, the study of the evolution of avalanche activity would require a specific focus on snow avalanche risk at the scale of a mountain area that is coherent in terms of climate and topography.

The contribution of historians to research on risk goes significantly beyond the simple production of large data sets from a variety of disparate sources. Their expertise is essential so that the occurrence of a natural phenomenon and its societal interpretation can be placed in their socio-historical and geographical contexts, so as to elucidate their meaning (Cœur and Lang, 2011). More broadly, contextualisation helps capture the temporality and dynamics of the various components involved in the risk system: biophysical factors (climate, hazard), practices and land use and vulnerability factors. The latter include stakeholder perceptions, representations of risk and the relation of societies to risk, as well as the capacity for society to deal with damageable phenomena. The historical approach is highly dependent on the existence of sources and, consequently, only possible if the area has been occupied for a long time. In cases where the usual sources are deficient, geo-

historical tools (Table 1) can also be utilised to complement standard historical methods. The purpose of this article is to demonstrate the value of such an approach for the (re)construction of a geo-chronology of avalanches and avalanche damage of sufficient duration and temporal resolution to understand the different characteristics of the phenomenon in the context of global change.

5  For this purpose, the Vosges Mountains is an ideal case study. Indeed, they combine topographical and snow features that are favourable to avalanche activity with long-lasting human occupation at relevant elevations, which should ensure the existence of appropriate data sources. After a detailed description of the geographic and socio-historical background, what follows describes the proposed methodology. It consists of an original combination of (re)sources of different types: administrative, printed, graphical, verbal and those resulting from spatial analyses. The results provided by this approach are

10  then presented, interpreted and discussed in light of the socio-historical, geographical and biophysical contexts of the region.

| Term | Definition |
|---|---|
| **Geo-historical approach** | This research approach mobilises historian's and geographers' questions, methods and tools. These participate in both the problematisation of the research object and its study. Within this framework, the notions of temporality, dynamics, spatiality, territoriality and social construction are combined. It allows understanding the at-risk system by integrating its spatial (perhaps territorial) dimensions as well as the temporality of the factors that are part of it. The at-risk system is explicitly considered as the result of a social construction. |
| **Geo-chronology** | Geo-chronology is related to a chronicle of events but goes beyond by putting them into their temporal and spatial contexts. |
| **(Re)sources** | This neologism relates to our corpus of documentation. It includes historical sources of various types (administrative, printed, graphical, verbal) and outcomes resulting from spatial analyses. Both allow identifying avalanches and/or contextualising them. |
| **Recorded avalanche** | 'Recorded avalanche' refers here to avalanches that have been identified. |
| **Event** | Events differ from physical occurrences of the hazard. For historians, the event is defined as 'what introduces a cut, a discontinuity (...)', in the 'everyday life' of individuals but also as what is 'interesting', seems sufficiently 'important' or 'new' to be 'told or put into action' (Dosse, 2010, Bertrand, 2010). This study focuses on all events that left a trace in the Vosges Mountains, in order to come as close as possible to the material reality of the avalanche phenomenon. In this sense, we define 'event' as all spatio-temporal occurrences of the avalanche phenomenon. We therefore consider avalanching as a non-ordinary phenomenon, normality being the non-occurrence of an avalanche. |
| **Event building** | The presence of a witness of the phenomenon or the discovery of the traces it left is an essential prerequisite, but this is not sufficient to result in an event. To reach this end, the observed facts should be considered to be sufficiently interesting to lead to the production of a narrative (written, oral), and, therefore, of a source. This process is called event building. As a result of a social construction, it is closely linked to the socio-historic context. It must finally be noted that event building is itself insufficient for an event to exist today. For this, its memory must have been transmitted. |

| | |
|---|---|
| **Traces** | The notion of trace refers to 'everything a phenomenon, in itself impossible to grasp [for past events], left' (Dosse, 2010). |
| **Source effect** | The source effect results from three interrelated factors that strongly interact during the study period: a diversification of the types of sources, an increase of the mass of documentation and a change in the event's emergence conditions. |

Table 1: Definition of few specific terms arising from the fields of history and social science and used in the present study.

## 2. Description of the territorial context

The Vosges Mountains are located in the north-east of France, in the western part of the Rhine Valley. They are generally associated with an imagery and landscape that contrast with the idea of massive avalanches: a mountain range on a human scale, described using a lexicon that conveys softness, accessibility and friendliness (Giacona *et al*., submitted a). Their dimensions are small, with a compact shape: 7300 km$^2$, 1423 m a.s.l. at its highest point, 150 km long with a width varying between 20 and 60 km. However, the southern part of the Vosges Mountains shows marked signs of Quaternary glaciation: U-shaped valleys, glacial cirques, moraine deposits and slopes greater than 30° (Flageollet, 2003). Oriented north-north-east – south-south-west, they form the first orographic barrier encountered by low pressure air flows coming from the Atlantic (Fig. 1). The cold and humid climate permits the accumulation of a deep snow cover. The latter persists in the form of snow patches until late spring and sometimes even until early summer in the glacial cirques located on the eastern slope. The proportion of snow precipitation to total precipitation is 20%, 30% and 60% at 700 m, 1000 m and 1350 m a.s.l., respectively (Wahl *et al*., 2009). The main ridge stands perpendicular to the prevailing winds that sweep its flat and bald apical surface. This loads snow on the eastern slopes, with impressive cornices forming at the break-of-slope (Fig. 1E). Such commonly produced wind slabs are a characteristic feature of the avalanche dynamics in the Vosges Mountains (Wahl *et al*., 2007).

In the Vosges Mountains, areas where avalanche activity is significant do not constitute permanent living spaces. However, human activity has been present in these areas for centuries (Kammerer, 2003) and several activities have left a durable imprint on the landscape. Among these, perhaps the most significant is the agro-forestry-pastoral system, which is a characteristic of the communities inhabiting the valleys. The harsh weather conditions and the seasonal snow cover limit the use of mountain pastures. Mountain farming therefore has two periods: summering at the higher elevations, between late May and late September, and wintering in the valley for the remainder of the year. The turn of the twentieth century marked the development of winter recreational activities, which resulted in a huge modification of the mountain calendar. Humans are now present in the uplands throughout the year. This in turn has had a concomitant impact on new processes of appropriation and mountain development. Examples are the extension of the network of trails for hiking, the construction of new buildings and the creation of ski resorts (Schwartz, 2001 and 2003).

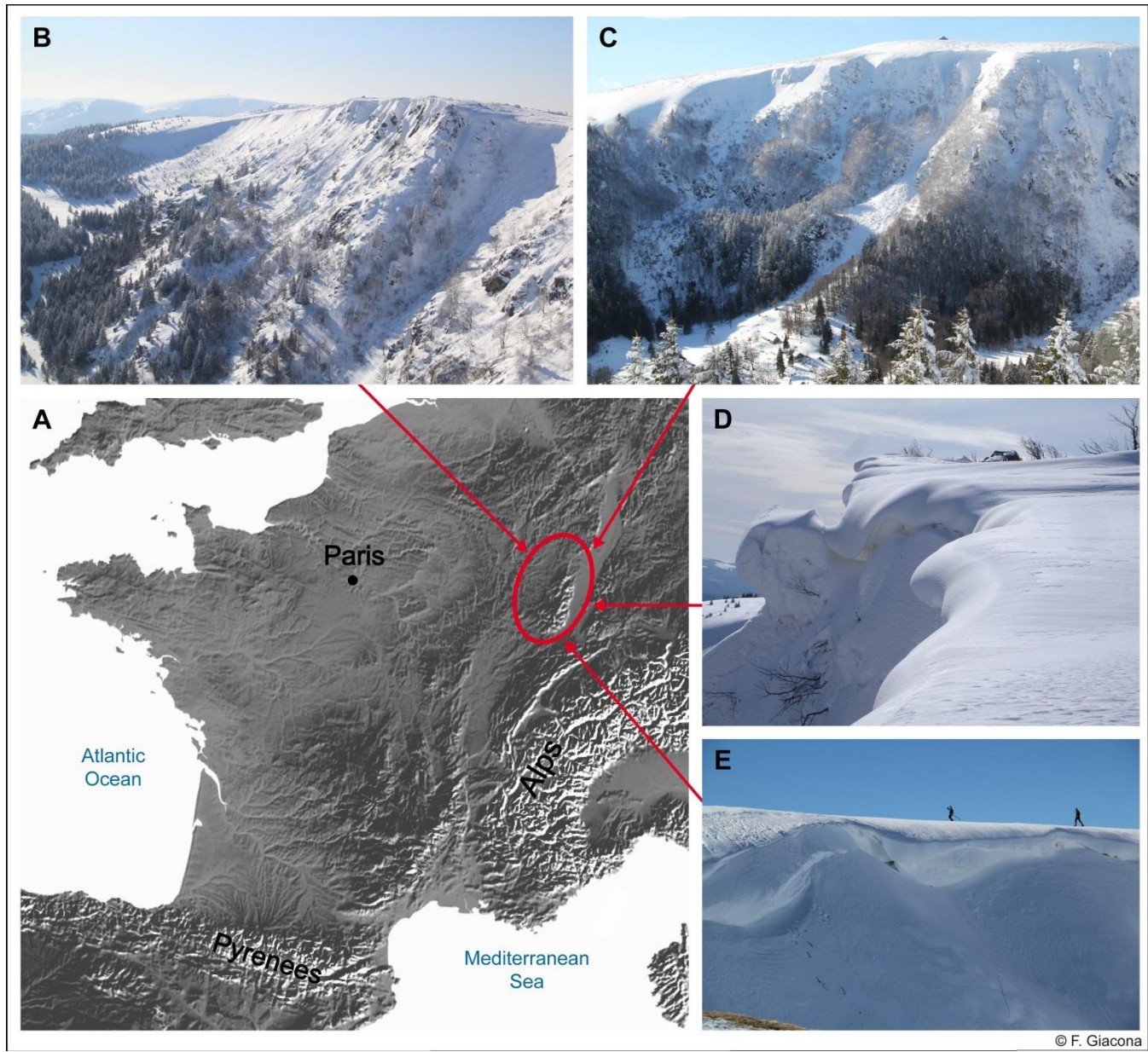

**Figure 1: Location of the Vosges Mountains in the north-east of France (A, red circle). Avalanches occur mainly in former glacial cirques (B, C). At the top of these, huge cornices often develop during the winter season (D, E), and their fracture is the most common avalanche triggering factor.**

5   Finally, the Vosges Mountains have never been, *a priori*, a barrier mountain (a mountain range with little communication across it, isolating one side from the other). However, its geographical location combined with its chaotic political history have been major obstacles to its perception as a complete and coherent entity and to the development of a real local memory and culture of natural hazards (Martin *et al*., 2015). This space has, in fact, been marked by several political breaks that have

concerned the Alsace and Alsace-Lorraine region over the period studied. Between 1870 and 1918, the Vosges Mountains were partitioned at their main ridge into two separate national territories. More generally, three Franco-German conflicts between 1870 and 1945 resulted in five changes in official language, nationality and administration. Despite these successive changes of national sovereignty, the last two centuries are characterised by continuity / consistency in terms of production and conservation of archives. Specifically, since the French Revolution, the different French departments have had an archivist and the production of documents relating to land use has continuously increased (Lang *et al*., 2003).

## 3. Geo-historical methodology

The local existing knowledge regarding the material reality of avalanche risk in the Vosges Mountains is low, with, *e.g.*, no systematic recording of activity, in contrast to the French Alps. This limitation combines with little information existing in institutional archives (at regional and municipal administrative scales). Indeed, only 15 avalanches over the last two centuries were found in church records, administrative documents, minute books of municipal council meetings, etc. This justifies the use of sources beyond the classical corpus the historian is used to working with. We have chosen to use a dual approach, integrating the analysis of traditional sources with field enquiries in both their physical (analysis of topography, vegetation etc.) and social meanings. Our social enquiries are based on surveys and interviews with people *a priori* concerned with avalanche risk in the Vosges Mountains: stakeholders, employees of resorts and ski schools, mountain professionals, participants of outdoor activities, members of history associations, plus various other users of the massif. Figure 2 summarises all the geo-historical tools used to expand the data corpus.

Specifically, browsing topographic maps of the French National Geographic Institute has helped to screen toponyms, but few of them refer to avalanches. Similarly, the scientific and regional literature reports few avalanche occurrences. Information relating in particular to avalanche accidents that have required rescue missions was more abundant in the regional media, especially in the press. To this was added the graphical material that could be collected: for instance, postcards and photographs (Fig. 3). The number of the latter illustrating avalanches increased from the last decade of the nineteenth century and exploded in the 1990's. This is a consequence of the increased human activity in the Vosges Mountains, and of the development of information and communication technologies.

The choice was also made to use oral testimonies of stakeholders involved in the Vosges Mountains and likely to be privy to pertinent information. These individuals made various contributions, ranging from bibliographic records to their own oral testimony of observed events. Finally, online forums and websites dedicated to backcountry activities in the Vosges Mountains were consulted, and a detailed enquiry was conducted. The latter focused on the risk cultures (knowledge and representations) of people participating in winter activities.

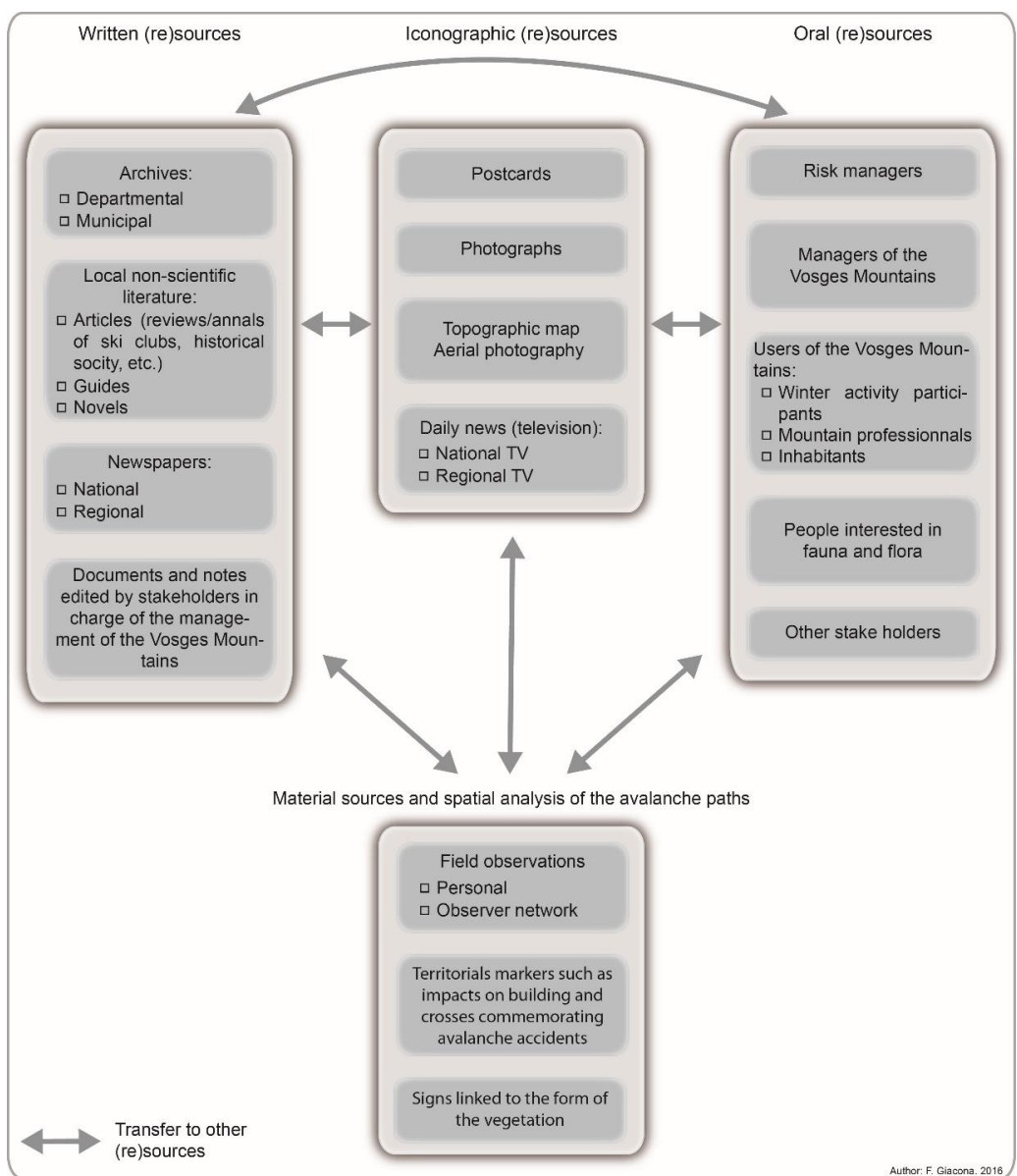

**Figure 2: Overview of the set of geo-historical (re)sources used in the present study.**

These different approaches allowed us to retrieve data on avalanches not mentioned in institutional archives. It is important to note the contribution of photographs, regional media, Internet sites and forums, as well as oral memory. These all aided in reconstructing a history of avalanches and enhanced their contextualisation (Table 2). Iconographic documents represent almost two-fifths of the total sources used. The proportion is slightly below one-fifth for oral testimonies and for regional media. A given source may refer to, from one to upward of 118 avalanches. However, four-fifths of the sources refer only to

a single avalanche. Moreover, a source may be contemporary with the avalanche or be produced later. Therefore, an avalanche may be identified by one or different sources (up to more than 60), possibly different in age.

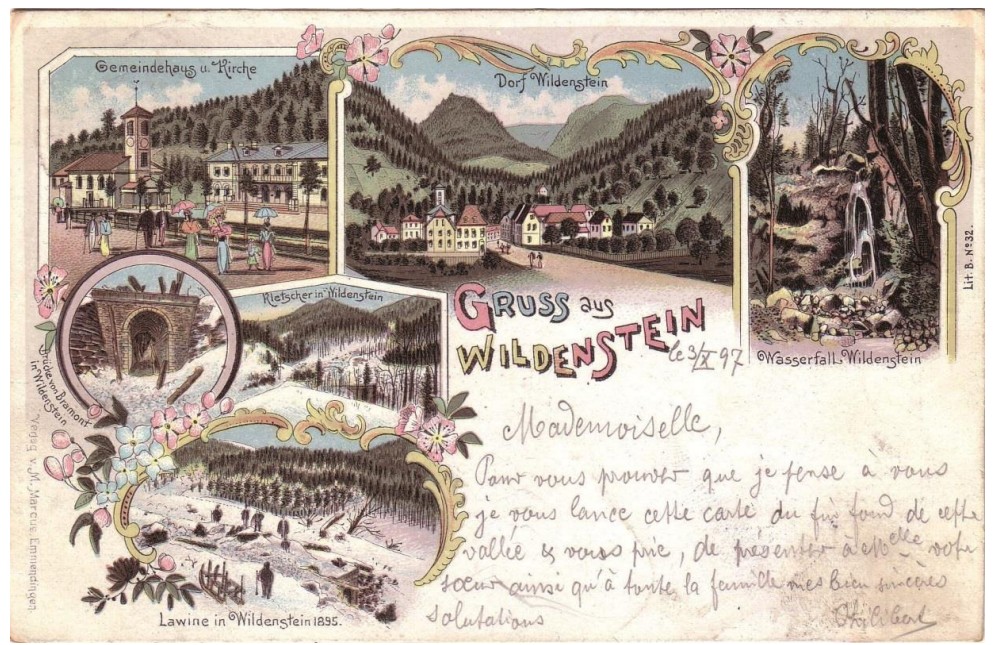

**Figure 3: Postcard of Wildenstein (township in the Vosges Mountains) that shows (bottom left) an avalanche that occurred in February 1895,** *Lawine* **meaning avalanche in German (Alsace was part of the 2nd Reich at that time). The postcard was circulated in 1897. Private Collection: J.-M. Ernst, Consultation through T. Meyer.**

| Type of sources | Number of documents / persons making references to one or more events |
|---|---|
| **Toponyms** | 1 |
| **Scientific literature** | 9 |
| **National media (newspaper and news programs)** | 10 |
| **Nonscientific and local literature** | 34 |
| **Departmental and municipal archives** | 58 |
| **Forums and web sites** | 135 |
| **Oral memory (testimonies, written correspondence, surveys, field observations)** | 189 |
| *Surveys* | *84* |
| **Regional and local media** | 198 |
| *Television (news programs and television magazines)* | *19* |
| *Newspapers* | *178* |
| **Images (postcard, photo)** | 417 |

**Table 2: Number of references to one or more events by type of source.**

These data were supplemented by diachronic spatial studies of avalanche paths. Direct field investigations including surveys of avalanche deposits and identification of various territorial markers (in particular, the presence of crosses commemorating a tragic avalanche) as well as discussions with observers (mountain professionals or winter activity participants) were conducted. These allowed us to locate the cornice formation areas, make a list of avalanche paths and collect information on past avalanche activity or on areas where past activity was likely.

All these (re)sources were sufficiently abundant and exploitable to reconstruct a geo-chronology. It includes a time series of past avalanches, a precise mapping of these under a GIS environment, and comprehensive information allowing the social and spatio-temporal contextualisation of the risk. The combination of questions, methods and tools from the fields of history and geography, as well as diachronic and multiscale approaches, facilitated the critical analysis of data and sources, and, more generally, the verification and cross-checking of all information. Three-fifths of the avalanches were corroborated by several sources. Among these, nearly two-thirds are known by two sources, and the remainder by at least three sources. However, it was not always possible to cross-check sources, in particular oral testimonies.

Avalanches were eventually classified according to different typologies. First, when possible, the intensity of each avalanche was rated on a specific intensity scale with five levels (from one: very low intensity to five: exceptional intensity). This scale is a function of the runout distance, the volume and spatial extent. It differs from the international avalanche intensity scale. Specifically, due to the size of avalanche paths in the Vosges Mountains, a level 5 on the international scale is impossible, and even a level 4 highly unlikely. However, an avalanche starting from the main ridge and flowing down to the valley bottom is, in the Vosges Mountains, a remarkable avalanche likely to have significant consequences, which warrants a level 5 quotation. Hence, the scale adapted to the physical characteristics of the Vosges Mountains reflects the variability of avalanche activity in the study area.

Second, avalanches having occasioned casualties, property damage, functional damage and environmental damage were highlighted. By casualties, we mean either people killed, injured and/or simply caught by the flow without any harmful consequences. Property damage refers mostly to partial or total destruction of one or several buildings. In rare cases, it also corresponds to damage to bridges or fences. Functional damage refers to traffic perturbation on roads. Environmental damage refers to disruption of ecosystems, up to the destruction of large forest stands (Table 3). Note finally that avalanche intensity and resulting damage are not synonymous, since low-intensity avalanches may sometimes have had substantial consequences. However, the two notions generally remain closely related. For example, major destruction of forest stands are always due to level 4 or 5 avalanches.

## 4. Results

### 4.1 Evolving sources over time

Quantitatively, the written and oral documentary mass related to avalanches is characterised by a considerable increase (almost exponential) during the period studied. The first change took place in the middle of the twentieth century, when the

number of sources was multiplied by three. Sources then increased by a factor of nine between the period from the 1940s to the early 1990s and the period covering the 1990s to today (Fig. 4C).

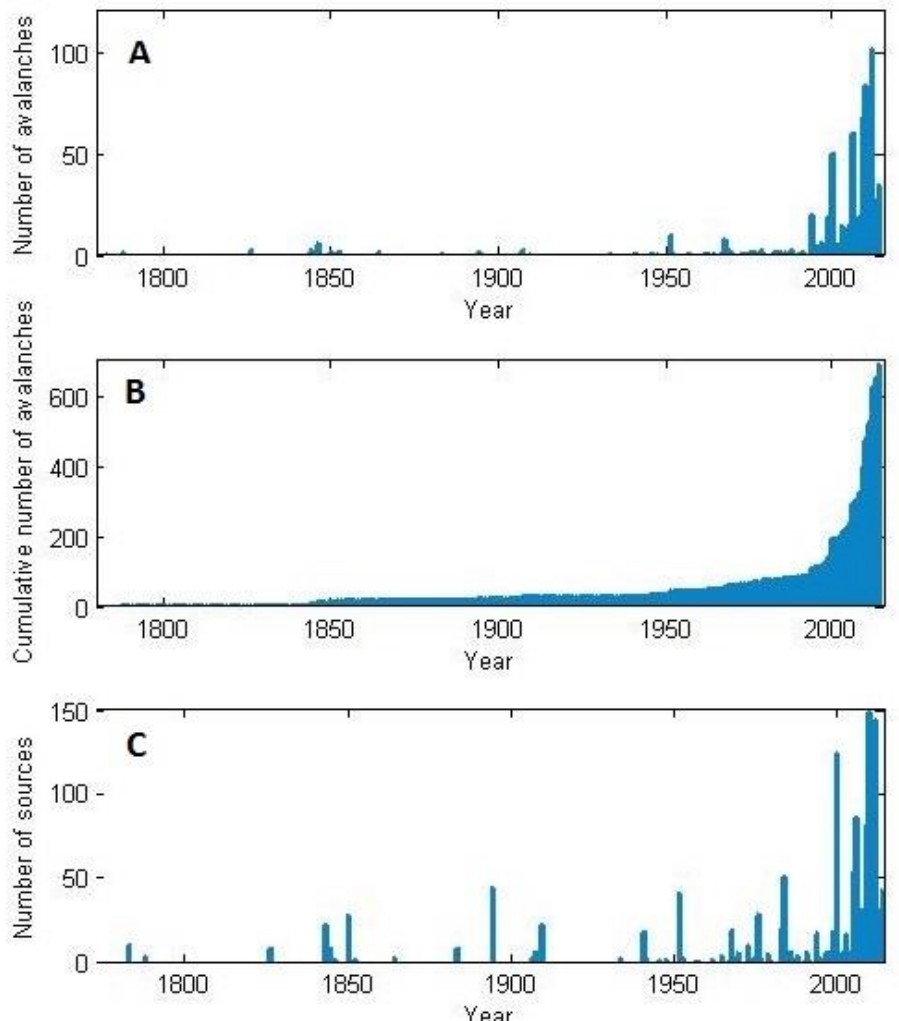

**Figure 4: Number of avalanches (A, B) and number of distinct sources (C) as a function of time. In A (number of avalanches per year) and B (cumulative plot), only the 682 avalanches with known year (winter) are considered.**

In addition to the non-homogeneity of the mass of documentation, a diversification of the nature of sources throughout the period also exists. Thus, five sub-periods were identified based on the predominance of different types of (re)sources available during the entire study period (Fig. 5).

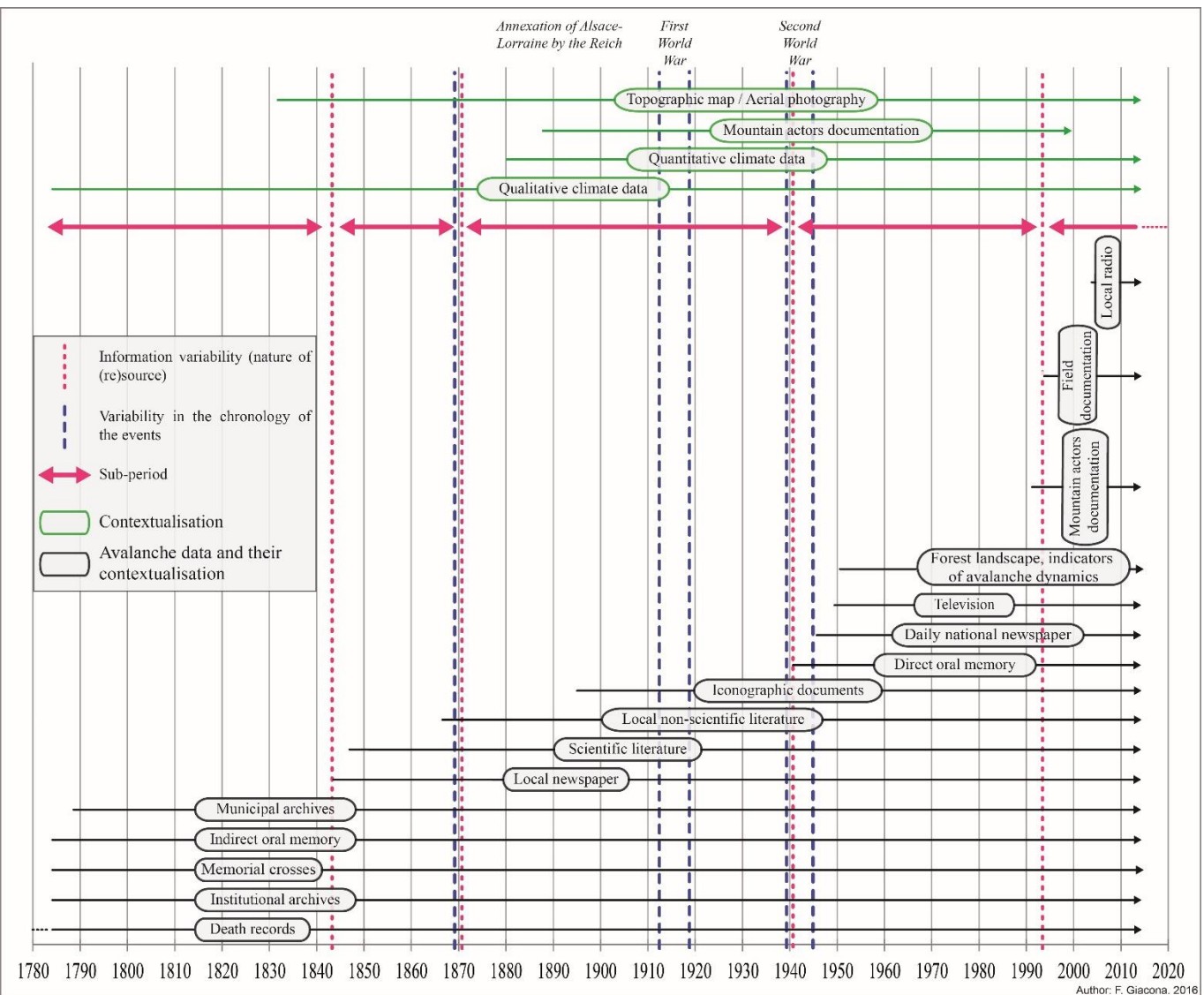

**Figure 5: (Re)source availability as a function of time. Sub-periods are delimited according to the predominance of a given type of source. 'Contextualisation sources' refers to sources that were used to describe the historical, geographical and societal context only, whereas 'avalanche data and their contextualisation sources' refers to sources that were used to both build the avalanche geo-chronology and understand the context of the corresponding occurrences.**

It should be noted that no (re)source type really disappeared. Whereas some of them did not continuously provide information, mentioning the occurrence of avalanches only occasionally, they were nevertheless considered as having been continuous from their appearance until today since their contribution could have been possible at any time. As available sources progressively became more numerous, one may have thought that this would have led to a diversification of information. However, in practice, this is not really the case, since each period is characterised by one or two main source(s):

- 1783/84–1842/43: essential contribution of administrative records;

- 1843/44–1869/70: first diversification, with contributions from the local literature and the regional press;
- 1870/71–1939/40: major contribution of the regional press and, to a lesser extent, of articles/scientific books and graphical materials that make their appearance. However, this period has to be considered in its political context. Several political breaks occurred: the annexation of Alsace-Lorraine to the German Reich in 1871, the First World War, the return of Alsace-Lorraine to France in 1918 and the beginning of World War II. These historical breaks had an impact on archives since the 1871–1918 and 1919–1945 periods are subject to a specific classification by the authorities (and some archival collections are incomplete, or have even disappeared);
- 1940/41–1992/93: predominance of the regional press, complemented by municipal archives, the local non-scientific literature, iconographic documents and news from regional television;
- 1992/93–2013/14: considerable contribution of graphic materials, followed by oral testimonies, forums and websites as well as the regional daily press. This period is an exception, since all the different types of sources are represented.

## 4.2 A spatially and temporally non homogeneous avalanche geo-chronology

The reconstructed geo-chronology includes 730 avalanches that occurred between the winters of 1783/84 and 2013/14 on 128 paths. Avalanches are therefore a common phenomenon in the Vosges Mountains. They have generated all kinds of damage: human, material, functional and environmental. The geo-historical approach also allows us to characterise their spatial and temporal distribution as well as their typology.

Over the entire period covered by this study, 94 avalanches caused one or more casualties. Among these, 16 were fatal to at least one person, with a total of 34 fatalities. Other casualties include injured and uninjured people. Seventeen avalanches have caused material damage, among which 11 led to the total destruction of one or several buildings (Table 3). In addition, 57 avalanches caused functional damage, mostly cutting off roads. Of these, 16 required the use of plowing machines. Finally, 80 avalanches damaged forest stands, among which 12 destroyed several hectares of forest. This typology of avalanches may contain some uncertainty. Indeed, the sources only contain information on the harmful nature of the event for half of the avalanches recorded. Nevertheless, it is reasonable to assume that most of the avalanches whose detailed impact is not known did not cause significant damage. Moreover, the data were used to classify nearly half of the avalanches according to their intensity. More than nine-tenths of the latter are of low to medium intensity, and less than one-tenth of them correspond to high to exceptional intensity.

Over the whole period, more than 90% of all the 730 avalanches identified (682 avalanches) could be specifically related to a given cold season (Table 3). The geo-chronology described below is based on these. The primary dating problem concerns recent decades, for which the information relates mainly to non-damaging and avalanches observed *a posteriori*. For such approximately dated avalanches (not very frequent), the uncertainty range is a few years maximum. By contrast, sources nearly always mention at least the year of occurrence (sometimes the exact day) for avalanches that have caused significant damage.

Locating the path of origin was possible for 520 of the 730 avalanches. All others, except one, were associated with a sector (a geographical area covering a few square km in which one to several avalanche paths are located). This allowed us to

determine the characteristics of a typical avalanche path in the Vosges Mountains (Table 4). Its length is close to 600 m, its average altitude 1100 m a.s.l., its mean slope 29° and vertical drop 240 m. However, the variability around these values is important, the path length varying, for example, from just under 40 m to more than 1350 m among the 128 paths identified for the Vosges Mountains. Also, the average altitude of the lowest path is around 440 m a.s.l., whereas the maximum vertical drop is close to 580 m.

| | Number of avalanches | Number of avalanches with known winter | % |
|---|---|---|---|
| **Total number of avalanches** | 730 | 682 | 93% |
| **Large avalanches (intensity level > 3)** | 28 | 26 | 93% |
| **Small avalanches (intensity level ≤ 3)** | 328 | 321 | 98% |
| **Unknown intensity level** | 374 | 335 | 90% |
| **With casualties** | 94 | 72 | 77% |
| **With people killed** | 16 | 16 | 100% |
| **With people injured** | 23 | 23 | 100% |
| **With property damage** | 17 | 17 | 100% |
| **With functional damage** | 57 | 54 | 95% |
| **With environmental damage** | 80 | 77 | 96% |

**Table 3: Typology of avalanches. Intensity level refers to a specific scale adapted to the characteristics of avalanche activity in the Vosges Mountains. By casualties, we mean either with people killed, injured or simply caught by the flow without any harmful consequences. Property damage refers mostly to partial or total destruction of one or several buildings. In rare cases, it also corresponds to damage to bridges or fences. Functional damage refers to traffic perturbation on roads. Environmental damage refers to disruption of ecosystems, up to the destruction of large forest stands.**

| | **Mean** | **SD** | **Minimum** | **Maximum** |
|---|---|---|---|---|
| **Length (m)** | 592 | 321 | 38 | 1351 |
| **Vertical drop (m)** | 241 | 105 | 5 | 576 |
| **Mean elevation (m a.s.l.)** | 1112 | 100 | 441 | 1336 |
| **Lowest point of the potential runout zone (m a.s.l.)** | 989 | 127 | 376 | 1257 |
| **Highest point of the potential release zone (m a.s.l.)** | 1231 | 99 | 469 | 1415 |
| **Mean slope (°)** | 29 | 5 | 16 | 42 |

**Table 4: Topographic characteristics of the avalanche paths where the 520 events that could be precisely located occurred. These were obtained by crossing the path GIS shape files with a 5-m resolution digital elevation model. The statistics take into account the specific activity of each path (when several avalanches, say *N*, occurred in the same path, the path's characteristics are weighted by a factor *N*.**

More than 95% of all avalanches occurred in the southern part of the Vosges Mountains where the elevation of the summits is significantly higher than in the rest of the mountain. Starting zones are mainly oriented to the north-east, east and the

south-east and, to a lesser extent, to the south. Very few are oriented to the west (Fig. 6). This is explained by the combination of factors favourable to the development of cornices and subsequent avalanches mentioned above.

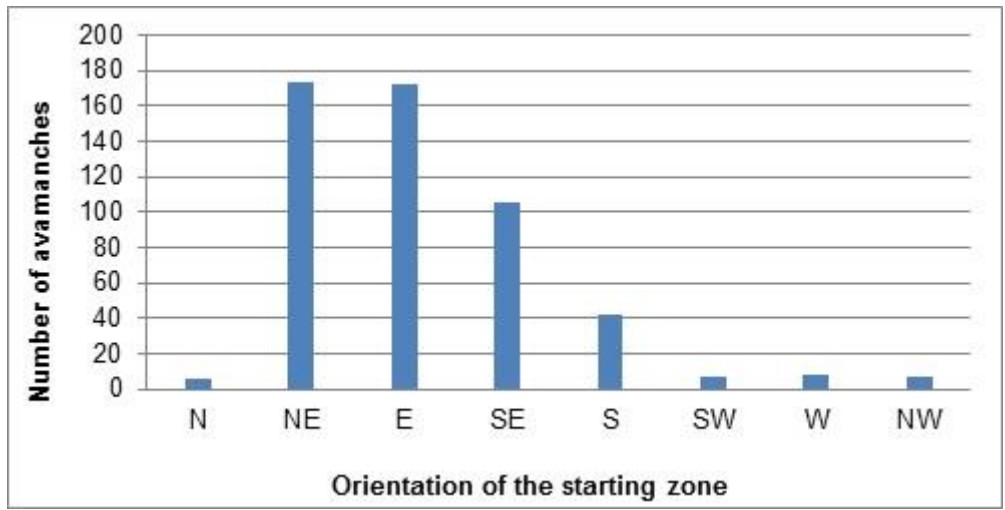

**Figure 6: Number of avalanches as a function of orientation of the starting zone. Only the 520 avalanches that could be precisely located are considered.**

The inventory clearly reveals specific temporal dynamics and singular rhythmicities. Thus, the raw chronology is marked by high temporal variability: the recorded avalanches remain sporadic until the mid-twentieth century (about 5% of all avalanches occurred prior to that date), with many winters having no avalanches. The number of avalanches per winter rises in the 1960s and increases even more in the 1990s (Fig. 4A-B). Nearly nine-tenths of the avalanches recorded occurred between the winters of 1989/90 and 2012/13, corresponding to the recent period during which there were no longer winters without any avalanches recorded.

Until the mid-twentieth century and even as late as the early 1990s, known avalanches were almost always high-intensity avalanches (intensity level greater than 3 on the scale previously introduced, Table 3). The difference between the pre-1990 records and the more complete recent records is that the number of low-intensity avalanches recorded (intensity level ≤3) drastically increased in the latter period (Fig. 7). Hence, the negative correlation between avalanche intensity and time is significant (Pearson correlation coefficient of −0.33 calculated from all avalanches for which the cold season and intensity are both available, significance assessed by a Student *t*-test).

Also, the nature of the damage caused by avalanches has changed over time. A few avalanches that resulted in significant damage are progressively outnumbered by more avalanches without significant consequences (Fig. 7 B–G). In the nineteenth century and the first half of the twentieth century, 15 avalanches which damaged or destroyed buildings were identified. Since then, no high-altitude farm or house located in the valley has been impacted. Some of these destructive avalanches have led to a significant number of deaths: up to ten during the destruction of a single building. Half of the avalanches that caused casualties have been recorded since the early 1990s. Over the same period almost two-thirds of the avalanches that have caused injuries were recorded. This recent increase is concomitant with a change in the nature of the victims impacted.

During the eighteenth and nineteenth centuries, the majority of the victims were people in their impacted homes, while in the twentieth century participants in backcountry activities are exclusively concerned. The mid-twentieth century also saw the emergence of a registry of injured victims, whereas before information was only retained, in general, for events that resulted in fatalities. Functional damage to transportation routes emerged in the middle of the twentieth century as mountain roads

5    were used increasingly in winter. Finally, the number of avalanches that caused damage to forests has also increased since the mid-twentieth century, but their relative proportion with regards to all recorded avalanches has decreased. The difference compared to other damage types such as the functional cases is that damage to forests by avalanches could be identified from the mid-nineteenth century.

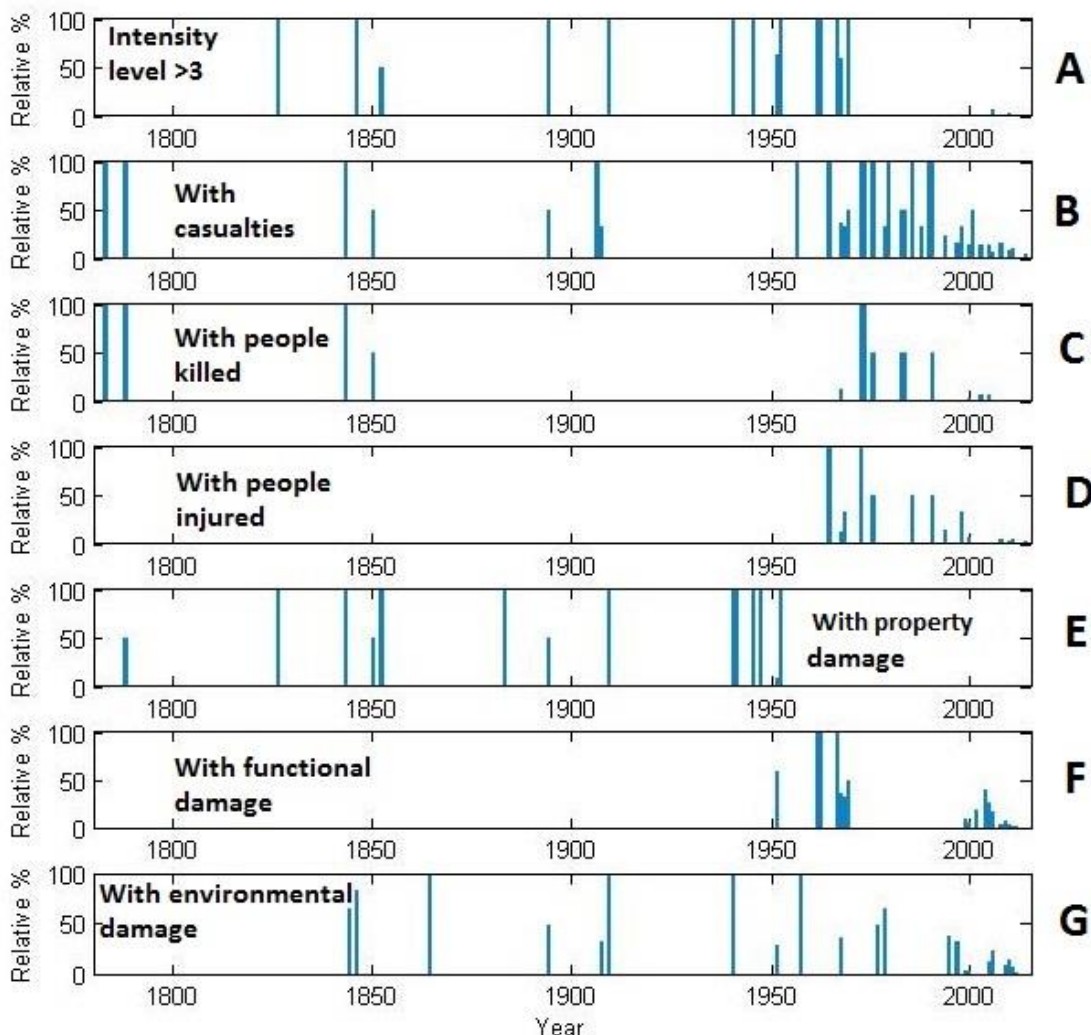

10    **Figure 7: Relative chronology of the types of avalanches recorded. A: Large avalanches with regards to all avalanches with a known size. B–F: Avalanches that caused damage: deaths, injuries, property damage, functional damage and environmental damage, with regards to all avalanches. All percentages refer to avalanches for which the year is known with certainty (Table 3, third column).**

Throughout the study period, there were 20 winters that were remarkable because of the number of victims and/or the extent of property or environmental damage. For example, in February 1844 the avalanche mentioned above caused the destruction of a house and led to the death of the ten people in Sainte-Marie-aux-Mines (Alsatian town located in a valley). Also, in February 1895, an avalanche starting from the Rothenbachkopf summit (located in the southern part of the Vosges Mountains) and flowing down the valley was sufficiently striking to be recorded on a postcard (Fig. 3). More recently, at the beginning of February 1952, three avalanches in the Rothenbachkopf-Rainkopf sector damaged 20 ha of forest, leaving within their deposit slightly more than 3000 m$^3$ of dead wood (Boithiot, 2001).

Finally, changes over time in the spatial distribution of avalanches is notable. Until the 1940s, the data relate mainly to paths located in valleys (Sainte-Marie-aux-Mines, for example) and on the main ridges of the mountains. From the mid-twentieth century and especially since 1993/94, the information concerns mostly paths located near or just below the summits (Rothenbachkopf-Rainkopf, for example) and there is a real extension of the spatial distribution of avalanche occurrences with the appearance of new avalanche paths in the geo-chronology. In addition, apart from a few exceptions, avalanches no longer occurred in the valley in this most recent period. The positive correlations between the cold season and the average elevation of the path, and, even more, between the cold season and the elevation of the starting zones (0.46 and 0.52, respectively, Student $t$-test, $p$-value <0.0001 in both cases) corroborate this hypothesis: recent avalanches have commonly occurred in paths that are, in the mean, located at higher elevations, and, even more significantly, have been triggered from starting zones located, on average, at higher elevations.

## 5. Discussion

### 5.1. How can the shape of the geo-chronology by explained?

To explain the shape of the geo-chronology, information must first be more precisely contextualised. Specifically, we consider that three factors, sometimes interdependent, can contribute to the interpretation of the periodicities highlighted: the sources effect, land use and avalanche activity.

### 5.1.1. Sources effect and event building

The number of avalanches follows more or less the same trend as the number of sources (correlation coefficient, 0.93 between the number of avalanches and distinct sources, $p$-value <0.0001). An ever-rising number of observations are recorded on an increasingly diverse range of media. Specifically, from the mid-twentieth century, avalanches were more regularly registered. This change is notably linked to the regional daily press, which relates accidents more frequently related to winter recreational activities (Fig. 4). Moreover, the clear increase in avalanches by cold season identified since 1993/94 is related to the contribution of the diary of a backcountry skier who has regularly recorded his field and weather condition observations since that season. A last significant development occurred in 2003/04 because since then, at least ten avalanches per winter season have occurred (Fig. 4A, B). This also marks the beginning of our own observations,

supplemented by those of the observer network, and information from forums and websites. The average increase in avalanche records, however, is not constant over the whole study period, with some years showing very few avalanches. This year-to-year variability can be partly explained by the irregularity of the sources, in combination or alone (depending on the cold season) with snow and meteorological conditions more or less favourable to avalanche activity.

Apart from the issue of the existence of potential occurrences, all observations must have left a trace as a written or/and oral documentary source. This process (that transforms observations into sources) has also varied, of course, over the period studied. Until the mid-twentieth century, this is conditioned by the occurrence of damage and is therefore closely related to vulnerability. Only avalanches that directly disturbed human lives have left a mark. From the second half of the twentieth century, recorded observations concern damaging as well as non-damaging avalanches, especially over the most recent
period (from 1993/94). This development is particularly related to guidance given to the observers' network to take into account all avalanches regardless of their consequence or their intensity level. The fact that people engaging in winter sports that feed the forums and websites are as interested in the phenomenon itself as its impacts also plays a role. Hence, in view of the large proportion of non-damaging avalanches recorded since 1993/94, as well as of the concomitant increase in small-sized avalanches recorded, we believe that avalanches recorded in earlier periods, that were generally harmful and often
large, account for only a small part of the avalanches that actually occurred. In the geo-chronology, damaging and large avalanches are thus over-represented before 1993/94 (Fig. 7).

All in all, the corpus of sources combines three types of information variability: the quantity of documentation, the type of available sources and the nature of avalanches identified the resulting source effect has a significant influence on the nature of the geo-chronology. Indeed, it is not because an avalanche occurs that the event exists. Production of a narrative (written, oral), and therefore of a source, following the observation of an avalanche and its potential damage defines an event
building. This is effective for a fraction of avalanches that actually occurred, according to criteria relating to the context. The 'building' of the event is the result of a social construction whose terms and rules have varied widely over the study period. In addition, of course, only part of the avalanches that have led to an event building appear in the geo-chronology, because some sources may have been missed by the historical investigation, while other events have been lost as a result of a lack of
transmission (end of oral memory, for example).

This gap between the reconstruction of avalanche activity from historical sources and field observations is illustrated by the comparison between the events of 1951/52 and 2009/10 in the Hohneck-Rothenbachkopf sector (Fig. 8). For winter 1951/52, we retrieved medium to exceptional intensity avalanches that caused functional, environmental and property damage, while no mention is made of smaller avalanches. On the other hand, for winter 2009/10, there is knowledge of many low-intensity
events, and of a few large-scale events with damage/size levels equivalent to those of 1951/52, and therefore likely to be retrieved from historical sources even decades later. By analogy, one may postulate that the events mentioned in historical sources for winter 1951/52 constitute part of the physical reality only, and that the actual situation of winter 1951/52 was similar to the winter of 2009/10.

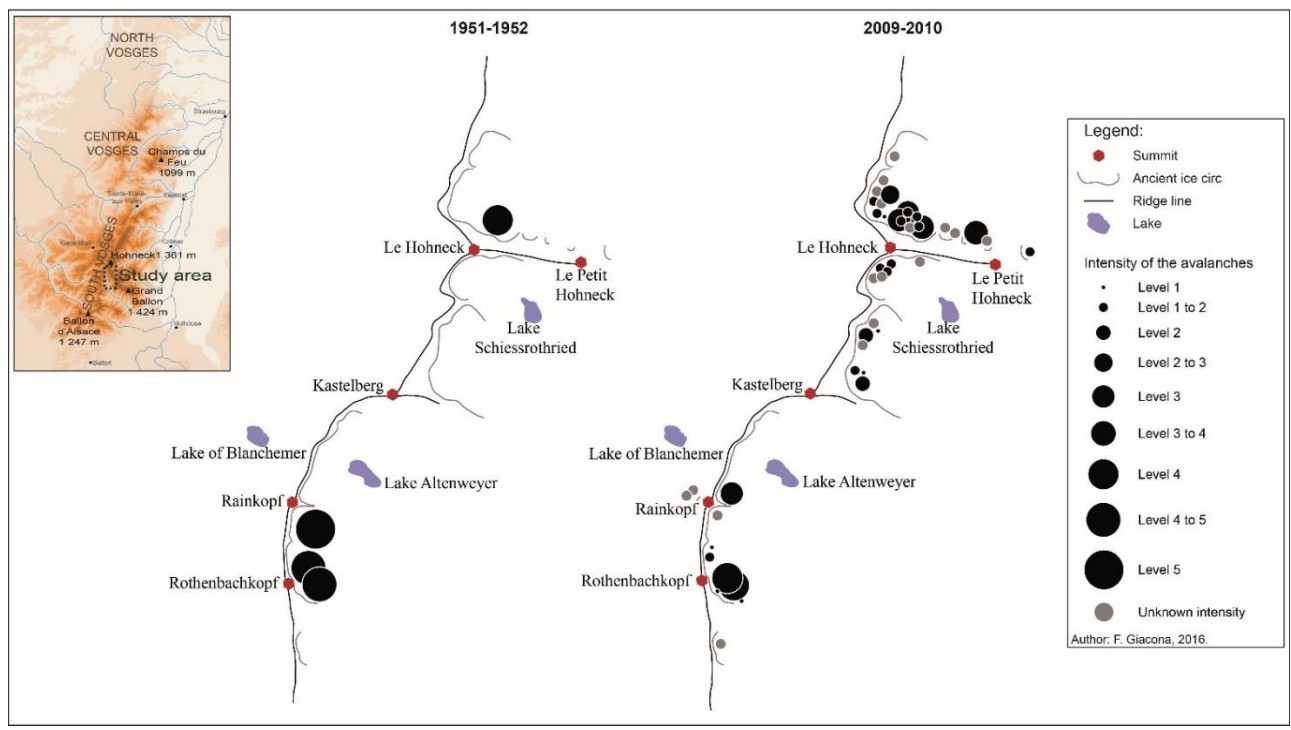

**Figure 8: Location and intensity level of the avalanches occurring in the Honneck-Rothenbachkopf sector during winters 1951/52 and 2009/10, respectively, obtained from the geo-chronology developed in this study.**

### 5.1.2. Land use changes

Land use in the Vosges Mountains has, both qualitatively and quantitatively, significantly changed over the 240-year study period. In the past, the Vosges Mountains were a living and resource area. Visits to the Vosges Mountains were mostly related to economic or family activities and exchanges, through the network of roads and paths connecting its various valleys. Merchants, smugglers, hawkers, hunters and poachers were their main users, without forgetting the loggers and *schlitteurs* (people who transported felled wood to the village with a *schlitte*, a kind of sledge transport). It is very difficult to

evaluate precisely which routes were used in the past and, especially, to assess how much each of them was used. However, we know that some of these roads crossed avalanche-prone areas. The record of accidents that occurred on tracks is an indicator of winter use of the Vosges Mountains. As an example, in March 1841, a brother and his sister were killed in a snowstorm on the stubble field of the Tanet while returning from an agriculture fair in Munster (Bresch, 1871). Archival documents also attest to the visit of high-altitude farms in the winter season. For instance, farmers were interested in

checking the status of their farms before the spring because if a damaged building was located less than a French half-mile from the forest, they had an obligation to ask the prefect for permission to rebuild or repair it (*e.g.* the letter of M. Mathias Guthleben to the prefect of the Haut-Rhin department, 24 March 1853, shelf mark 7P638, Departmental archives of the Haut-Rhin). Indeed, a regional book mentions the need to repair upland farms in the spring season 'if the farm or the barn is

exposed to winds from the west or to the devastation of avalanches, the roof needs to be repaired, the overthought risen, the mountain path repaired and cleared from the stones that have rolled onto it' (Abel, 1913).

Hence, the low abundance of sources up to the mid-twentieth century is not explained by the absence of stakeholder activity in these regions. The reason is presumably rather to be sought in the connections that previous societies had to the hazard and to the risk in the Vosges Mountains. Unfortunately, these remain difficult to grasp, due to incomplete sources, and, more generally, because mountain people were characterised by an oral culture rather than a written culture. As stated above, we know that upland farmers certainly identified an avalanche risk, at least as early as the nineteenth century. Having traveled, mapped and modified this territory, they undoubtedly developed knowledge resulting from their experience of this space, and perhaps also transmitted knowledge and know-how. On the other hand, it is not certain that the phenomenon fundamentally influenced their practices and techniques, induced particularly elaborate precautionary principles, or even that there was real comprehension of the threat. The main reason is that avalanches only rarely affected permanent living places. Indeed, even for upland farmers, the choice to spend the entire year in the Vosges Mountains was an exception. Hence, avalanches have not induced structural damage to the mechanisms of the territorial system – physical or socio-economic – especially with regards to other social risks such as epidemics or armed conflicts that had far greater consequences and, as a result, presumably structured the history and life of mountain societies in a much more significant manner. This presumably explains why the records do not show attention paid to avalanches "sufficient to include them regularly in newspapers, private and local archives" (Granet-Abisset, 2012), so that the occurrence of damaging avalanches led only to mandatory documents required by the administration. It also explains why references to past avalanches remain limited to a few facts, without further detailed explanation or descriptive indication, and that the memory of the risk remained for the "short time of the event, experienced during a lifetime" (Barrué-Pastor, 2014). All in all, current knowledge therefore does not result from an inheritance or oral transmission of experience from one generation to another.

Eventually, it is clear that today the situation has changed dramatically. The Vosges Mountains have become an area mainly dedicated to leisure for a wider population. Even in the winter season, the mountain is visited entirely (including avalanche-prone areas) for recreational purposes. It is difficult to assess how visits to the backcountry sectors have evolved. Yet, it is certain that this is growing, as attested by the rising number of avalanche accidents in these areas. This increase in visits partly explains the rise of avalanches recorded from the mid-twentieth century and, to a greater extent, since 1990 (Fig. 6A). Indeed, it has extended the spatial coverage of information (with the registration of avalanches in paths located below the summits where no permanent property or functional items at risk exist) since the mid-twentieth century (Fig. 4B), and even more since the 1990's. This explains the new spatial distribution of avalanche accidents over the recent years, and, more widely, observations of natural avalanches that were missed before.

Finally, in addition to an increase of potential sources (direct observation, even in the most remote part of the mountains), there is an increase in artificially triggered avalanches (as opposed to spontaneous avalanches) resulting from an overload at a specific time, caused by the passage of an individual for example. Consequently, people are playing a more active role in triggering avalanches, directly exacerbating the hazard. Meanwhile, unlike human vulnerability, the exposure of property at

risk (high-altitude farms in particular) has decreased significantly following the progressive abandonment of agricultural activities. In the language of risk analysis, rather than an emergence of a natural hazard in areas where it did not exist before, human exposure to the natural hazard itself has increased, which in turn has contributed to a greater visibility of the natural phenomenon.

### 5.1.3. Evolving avalanche activity under climate and land cover changes

This study takes place in a well-defined climatic context, the recent anthropogenic warming over the last few decades and the end of the Little Ice Age in the second half of the nineteenth century (Grove, 1988; 2001). The exact effect of such changes on the spatio-temporal distribution of recorded events is difficult to determine (Eckert *et al.*, 2010; Castebrunet *et al.*, 2012). However, colder winter temperatures until the middle of the nineteenth century enabled the development of an

10 important snowpack at medium (~900 m a.s.l.) and low (~600 m a.s.l.) altitudes. This has become exceptional today due to warming. It is therefore *a priori* reasonable that natural avalanche activity has generally declined over the study period, at least on the scale of the Vosges Mountains (at the local level, examples of paths where activity has risen may be found due to the specificity of each path). This evolution, which is in contradiction with the overall shape of the geo-chronology, confirms the preponderance of the social significance of the event.

The climate also indirectly impacts avalanche activity through its effect on afforestation and land use. More broadly, strong interactions between avalanches, forest and society exist (Bebi *et al.*, 2009; Feistl *et al.*, 2015). Pressure on wood resources remained strong until the nineteenth century, leading to the extension of deforested areas (including pastures) at the expense of forested areas, and therefore, presumably, to an intensification of avalanche activity. It therefore seems possible to link the destruction by avalanches, in the nineteenth century or early twentieth century, of upland farms or houses located in valleys,

with deforestation carried out, especially for grazing, combined with exceptionally harsh winter conditions. Since then, the context of climate warming, combined with the abandonment of pastures and proto-industrial activities, and man-made reforestation from the early nineteenth century have favoured forest recolonisation, and, presumably, decreasing avalanche activity. As a consequence, some paths in the valley areas, where in the nineteenth century destructive avalanches occurred, have disappeared. Indeed, they are now fully re-colonised by forest, and no longer show any signs of avalanche activity.

Note, however, that pastoral activity is still very strong on the high summit pastures. This hinders forest recolonisation and promotes sweeping snow on flat ridges and thus the formation of wind slabs and imposing cornices on the eastern slope, which remain the most active in terms of avalanche activity (Fig. 1).

Finally, recreational activities, and therefore winter use of the Vosges Mountains, are partly based on the state of the snowpack. The absence or scarcity of snow is indeed generally synonymous with less intense backcountry ski activities, and

30 therefore to less accidental triggering. All factors are therefore thoroughly interconnected.

## 5.2. Comparison with other contexts

In France, the avalanche database EPA covers the Alps and the Pyrenees. It includes nearly 100,000 entries over a little more than one century (Bourova *et al*., 2016). Similarly, many countries exposed to avalanche risk regularly register avalanche activity (Bonnefoy-Demongeot *et al*., 2014). For example, in Switzerland, avalanche data have been systematically collected since 1950 by the WSL Institute for Snow and Avalanche Research SLF (Techel and Zweifel, 2013) and, in the United States, a generalised long-term database of weather, snowpack and avalanche information has been available since 1967 (Williams, 1994). Compared to such inventories, our geo-chronology is not very rich at first glance. However, raw numbers should be compared with care, since they depend on the size of the area considered. More than 700 events for a small range like the Vosges Mountains is considerable, and the unusually long time frame covered makes this inventory even more valuable. As a comparison, Podolskyi *et al*. (2014) reports not more than 275 events for the whole Sakhalin Island over the 1910–2010 period. Also, contrary to ours, most of the existing national inventories document only the last a few decades.

Furthermore, as stated above, in many countries/areas, such records simply do not exist. For instance, even if France clearly stands among the countries where documentation of past avalanches has been the most systematic, certain parts of the territory have been completely missed so far, namely medium-high mountain ranges. This concept of medium-high mountains remains highly subjective, especially in terms of physical and social criteria. In France, it refers roughly to areas ranging from 600–700 to 2000 m. a.s.l. These are often contrasted to high-altitude mountain areas (the Alps and Pyrenees), in terms of altitude, topography (old versus young, rounded tops versus rugged peaks, etc.), natural processes (seasonal snow cover only versus glaciers) or practices (everyday living space versus marked seasonal rhythm associated with tourism). Medium-high mountains are finally characterised by the combination of a humanised mountain (everyday living space and possessing a rich cultural heritage) and preserved wilderness areas (Sgard *et al*., 2007; Giacona *et al*., submitted a). Beyond France, it might be fair to say that, apart from exceptions (*e.g.* Hétu *et al*., 2011), such mountain ranges have been largely overlooked with regards to high mountain ranges (Andes Cordillera, Rocky Mountains, European Alps, etc.).

In terms of risk and accident tolls also, this chronicle is stronger than at first glance. While there has been a total of ten avalanche deaths in the Vosges Mountains due to recreational activities since the winter of 1971/72, the national average in France is around 30 deaths per year (Jarry, 2011), and approximately 100 deaths per year throughout the Alps covered by the recent inventory of Techel *et al*. (2016). The physical characteristics of avalanche activity (altitude, soil type in the avalanche path, snow climatology, size, intensity and frequency of events, etc.) and vulnerability are undoubtedly different in the Vosges Mountains and in other mountain areas such as the Alps. However, at the regional level, the ten deaths in the Vosges Mountains are far from negligible. Indeed, it is not certain that, at the individual level of the skier or the mountain hiker, the risk is significantly lower in the Vosges Mountains than in higher, larger mountain ranges.

Finally, differences and similarities with other contexts in terms of risk perception and culture can be noted. In the Vosges Mountains, the risk was not part of everyday life and upland farmers did not have to really learn to live with it. They probably did not develop specific territorialised knowledge 'with meter-accuracy', as may have been the case in the Alpine

and Pyrenean mountain areas where permanent living places were threatened during the winter season (Barrué-Pastor, 2014; Granet-Abisset, 2012). This more generally means that the constraints for past societies due to avalanches were different in the Vosges Mountains than in high-mountain areas. On the other hand, a low abundance of written (re)sources can be encountered in other mountain areas, including high mountain ones. Barrué-Pastor (2014) notes for example the existence of

a mountain risk culture in the Pays Toy, an area in the French Pyrenees, including avalanches, but one that has left no written record. The risk was in fact apprehended through stories, architecture and toponyms. These examples illustrate how interactions between avalanche activity and society vary across physical and societal contexts. The analysis could be largely expanded in further work. However, the current lack of long reconstructions of contextualised chronicles of avalanche events and risk in different contexts clearly stands for now as a limitation to this appealing perspective.

**6. Conclusion and outlooks**

This study shows that it is possible to reconstruct *a posteriori* a geo-chronology of avalanches, even for a relatively benign medium-high mountain area that one would barely think of as prone to avalanche risk. This requires the combination of the historian and geographer's questions, methods and tools, which have undeniable strongpoints: specifically, diachronic and multiscale approaches applied to conventional, less common (re)sources allow building a comprehensive geo-chronology by

looking for facts, cross-checking them and analysing and retrieving their meaning in the social space. It then becomes possible to understand its shape, namely singular periodicities and changes resulting from interactions between the temporality of social actors and nature. The importance of the work done (variety and volume of the relevant (re)sources studied, extensive field visits conducted, etc.) as well as the consistency of the chosen spatial scale (a whole mountain range) give confidence in the robustness of our conclusions. This approach is very innovative in the field of natural hazards,

particularly avalanches, where existing historical studies remain relatively rare, and often based on information less widely collected and/or considered at a spatial scale less relevant for the physical process.

To challenge the way the society relates to the avalanche phenomenon and, more generally, how the at-risk system is generated, the analysis exploits very diverse kinds of data. However, a generic approach is constantly used, namely information contextualisation, without which results are uninterpretable. This is a classic but relevant demonstration of the

fundamental role of the historian in the study of risks over a long time period. Although the results obtained are highly dependent on the geographical and institutional contexts, on the local socio-environmental relations (land use, memory, risk representations, relation to the risk, etc.) and on their interactions, the overall reflection and the methodology used could be profitably transposed to other risks or to other mountain areas. This applies particularly to medium-high mountain regions where knowledge is still very partial, if not absent.

The results show that avalanche activity and risk are significant in the Vosges Mountains, both in terms of frequency and intensity. Avalanches have caused a dozen deaths since 1970, which make it one of the deadliest natural hazards in the Alsace region. It therefore seems reasonable to think that the low level of avalanche risk warning in the Vosges Mountains is

more due to a lack of consideration of this phenomenon as a public problem than it being related to an actual absence of avalanche activity. This low visibility of avalanche activity and risk is explained historically and socially by the local relations to the avalanche phenomenon we have detailed. With few exceptions, the risk does not constitute a structural constraint, neither for past and present everyday life, nor for the activities in the Vosges Mountains. The risk has been and

5 remains essentially to individuals. In this context, it is not surprising that it does not appear as a component of the local mountain culture. For instance, little evidence of collective heritage could be identified, except the existence of a few markers such as crosses, a postcard and local stories.

The geo-chronology we have built reflects only part of the avalanches that actually occurred during the study period. The low number of avalanches recorded until the mid-twentieth century essentially corresponds to damaging avalanches,

especially large ones, whereas more recent avalanches include many more events not resulting in damage. This is because our geo-chronology was built to be as close as possible to the reality of avalanches in the Vosges Mountains, which means that it includes both damageable and non-damageable events. As a consequence, it is neither a chronicle of avalanche hazard nor a chronicle of avalanche risk, but it contains information relevant for both. Rather than a detailed investigation, only a rapid analysis of this abundant information was performed. Hence, further research could clearly exploit it in many

directions including avalanche-climate inference, flow modelling, quantification of risk changes over time, etc.

As an unavoidable preliminary work for such appealing outlooks, we focused on the contextualisation step. The inhomogeneity of the timeline could not be explained solely through the corpus of (re)sources because of the importance of geographical and biophysical contexts. Indeed, the increasing human pressure and deforestation during the eighteenth century, in a context probably aggravated by large amounts of snow during the Little Ice Age, may have favoured the

20 formation of large-scale avalanches but also allowed activity in low-altitude paths located close to valley bottoms. In contrast, combined man-made and natural reforestation then led to a reduction in avalanche frequency and intensity, and even to the disappearance of avalanches in certain valley sites. This is attested to by instructions given to scientists and managers today to remove trees that grow in certain areas, no longer hampered by avalanche activity, in order to preserve the natural plant and animal heritage typical of the Vosges Mountains landscape (Conservatoire des sites lorrains and Service

d'appui technique Office national des forêts de Colmar, 1999). Therefore, the analysis shows that the temporal dynamics of the hazard is not only influenced by natural factors, but also depends heavily on interactions between society and its environment, both through human impacts on the environment and through the presence of stakes that may play a role in the triggering mechanisms.

More generally, because knowledge of its occurrence is partially conditioned by the (re)sources available, the avalanche

exists only as an object perceived and constructed by societies. The event is a social construction, related to the relations between the society and its environment. Consequently, the shape of the geo-chronology reflects the complex interrelationships between various factors: i) the (re)sources, which are dependent on events built from the facts, and on their transmission, ii) the social practices of the territory conditioning vulnerability, from which the production of sources is partially dependent, and iii) the natural phenomenon resulting from a combination of meteorological and topographical

characteristics with triggering factors. The reproduction of this type of study in other contexts should lead to a better understanding of this rich but complex system in the future.

## 7. Acknowledgements

We sincerely thank C. J. Keylock for proof-reading the manuscript, and the two anonymous referees whose detailed comments helped improve the clarity of the paper's content. The many people who contributed in various ways to the constitution of our avalanche geo-chronology are also acknowledged.

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
