# Peer review of "A 240-Year History of Avalanche Risk in the Vosges Mountains Based on Nonconventional Sources"

_Natural Hazards and Earth System Sciences, 2016_

## Referee Comment (RC1) · Anonymous Referee #1 · 18 Jan 2017

Dear Editor, dear Authors,

This contribution covers a very important and interesting topic, namely the search, analysis and exploitation of conventional and nonconventional information sources to establish and/or extend databases of past natural hazard events. The article points out the many challenges associated with data acquisition over a long study period during which information sources evolved substantially. The text is generally well illustrated and the overall structure of the manuscript is adequate.

Data on natural hazard processes (especially when destructive) are used by a wide variety of organisations (e.g. scientific institutes, private environment companies, insurance companies, governments). Such data are a very useful instrument in providing basic information for better hazard and risk assessment as well as decision-making. It

is therefore important to promote and support studies such as the one presented in this manuscript.

The case study presented by the authors focuses on a single hazard type (snow avalanches) and on regional issues (Massif des Vosges). It is in many ways special which partly makes it even more valuable. Most importantly, (i) it covers a very long time frame, (ii) it covers an area not especially known for the hazard process of snow avalanches and (iii) it treats an area that has experienced fundamental turbulences in the past (e.g. several armed conflicts in the last 150 years; changes in official language etc.). These points reveal methodological research problems that one may not encounter in similar investigations elsewhere.

Given the importance of natural hazard event analyses and data sets, I think this valuable contribution should be published and I hope that it will motivate researches in other regions and dealing with other hazard processes to follow this (obviously quite demanding) path.

However (!), I see several substantial (mainly formal and structural) problems that need to be solved before this text is ready for publication in NHESS.

(i) First of all, the use of English language is not very good and needs to be substantially improved. In many places, the constructed sentences are much too long and contain too much information that make them often confusing and difficult to understand. The authors should try to formulate short, reader-friendly and clear sentences. I tried to make suggestions/corrections where possible. However, because I am not a native-English speaker my proofÂňreading is not at all complete. I am strongly convinced that the text would benefit from thorough editing by a native-English speaker.

(ii) Footnotes are used throughout the manuscript. I would strongly recommend to avoid the use of such footnotes. It is not, to the best of my knowledge, acceptable in this journal. The contents of many of these footnotes is not essential for the understanding of the text. They could e.g. be summarised in a supplementary file associated to the

article or could partly be deleted. The most important contents should be incorporated in the main text or in figure captions.

(iii) In my opinion, the manuscript is currently too long. A lot of the issues addressed in the text are described in too much detail. In some parts (e.g. Discussion), the text is repetitive. I think it is crucial that the authors rework the manuscript, mainly sections 4 and 5, by carefully picking the main statements they want to make. According their decision, these issues and statements need to be accurately put to paper in simple but significant sentences (see also first point above).

(iv) Some key terms used throughout the manuscript should probably be introduced and precisely be defined in the Methods section. For example, the use of the term event (also source event, historical event, observed events, avalanche event etc.) is not clear and confusing at times. At page18/line15 the authors state that "it is not because an avalanche occurs that the event exists" and then produce a definition from literature. This is really complicated to understand and should (in my opinion) be clarified earlier in the text.

In summary, I do not think that this manuscript is ready for publication yet. I suggest the authors revise their text and solve the formal problems mentioned above. As regards content, the article is on a good level. However, the information the authors want to communicate in the latter sections of the manuscript should be reassessed, reorganized and if possible shortened. I do sincerely think that there is potential for an NHESS article in this manuscript. I thus recommend that the paper be accepted pending major revisions or that it be rejected with an invitation to re-submit when all formal aspects criticised are clarified. I provide a separate list of partly detailed comments specific to the different sections of the article as well as to the tables and figures produced by the authors. Additionally, I prepared a list of technical corrections for the authors (same separate document).

Please also note the supplement to this comment:

[Figure]

http://www.nat-hazards-earth-syst-sci-discuss.net/nhess-2016-395/nhess-2016-395-RC1-supplement.pdf

[Figure]

**Supplement:**

**Review of manuscript NHESS-2016-395:**

**"A 240 year History of Avalanche Risk in the Vosges Mountains from Nonconventional Sources"**

**by Giacona F., Eckert N., and Martin B.**

**1) General comments**

Dear Editor, dear Authors,

This contribution covers a very important and interesting topic, namely the search, analysis and exploitation of conventional and nonconventional information sources to establish and/or extend databases of past natural hazard events. The article points out the many challenges associated with data acquisition over a long study period during which information sources evolved substantially. The text is generally well illustrated and the overall structure of the manuscript is adequate.

Data on natural hazard processes (especially when destructive) are used by a wide variety of organisations (e.g. scientific institutes, private environment companies, insurance companies, governments). Such data are a very useful instrument in providing basic information for better hazard and risk assessment as well as decision-making. It is therefore important to promote and support studies such as the one presented in this manuscript.

The case study presented by the authors focuses on a single hazard type (snow avalanches) and on regional issues (Massif des Vosges). It is in many ways special which partly makes it even more valuable. Most importantly, (i) it covers a very long time frame, (ii) it covers an area not especially known for the hazard process of snow avalanches and (iii) it treats an area that has experienced fundamental turbulences in the past (e.g. several armed conflicts in the last 150 years; changes in official language etc.). These points reveal methodological research problems that one may not encounter in similar investigations elsewhere.

Given the importance of natural hazard event analyses and data sets, I think this valuable contribution should be published and I hope that it will motivate researches in other regions and dealing with other hazard processes to follow this (obviously quite demanding) path.

However (!), I see several substantial (mainly formal and structural) problems that need to be solved before this text is ready for publication in NHESS.

- First of all, the use of English language is not very good and needs to be substantially improved. In many places, the constructed sentences are much too long and contain too much information that make them often confusing and difficult to understand. The authors should try to formulate short, reader-friendly and clear sentences. I tried to make suggestions/corrections where possible. However, because I am not a native-English speaker my proofreading is not at all

complete. I am strongly convinced that the text would benefit from thorough editing by a native-English speaker.

- Footnotes are used throughout the manuscript. I would strongly recommend to avoid the use of such footnotes. It is not, to the best of my knowledge, acceptable in this journal. The contents of many of these footnotes is not essential for the understanding of the text. They could e.g. be summarised in a supplementary file associated to the article or could partly be deleted. The most important contents should be incorporated in the main text or in figure captions.

- In my opinion, the manuscript is currently too long. A lot of the issues addressed in the text are described in too much detail. In some parts (e.g. Discussion), the text is repetitive. I think it is crucial that the authors rework the manuscript, mainly sections 4 and 5, by carefully picking the main statements they want to make. According their decision, these issues and statements need to be accurately put to paper in simple but significant sentences (see also first point above).

- Some key terms used throughout the manuscript should probably be introduced and precisely be defined in the Methods section.
  For example, the use of the term *event* (also *source event*, *historical event*, *observed events*, *avalanche event* etc.) is not clear and confusing at times. At page18/line15 the authors state that "it is not because an avalanche occurs that the event exists" and then produce a definition from literature. This is really complicated to understand and should (in my opinion) be clarified earlier in the text.

In summary, I do not think that this manuscript is ready for publication yet. I suggest the authors revise their text and solve the formal problems mentioned above. As regards content, the article is on a good level. However, the information the authors want to communicate in the latter sections of the manuscript should be reassessed, reorganized and if possible shortened.

I do sincerely think that there is potential for an NHESS article in this manuscript. I thus recommend that the paper be accepted pending major revisions or that it be rejected with an invitation to re-submit when all formal aspects criticised are clarified. I provide below a list of partly detailed comments specific to the different sections of the article as well as to the tables and figures produced by the authors. Additionally, I prepared a list of technical corrections for the authors.

**2) Specific comments regarding the different sections of the article**

**Abstract**

The Abstract is well structured and of adequate length.

P1-L9/10:    Why the use of "still"? Consider changing to:
             "especially in medium-high mountain ranges with a significant process
             activity."

P1-L11:      A real problem of this MS is the lack of consistency in the spelling of terms.
             A good example is the term "north-east of France" at line 11 that throughout
             the text is also spelled "North-East of France" and "North East of France".

P1-L12:      Same point: "geo-chronology" is also spelled "geochronology" in the MS. Be
             consistent.

P1-L17/18:   I do not understand what is meant by "for risk changes understanding and
             mitigation"

**Introduction**

The Introduction is not so well organized. Towards the end of it the authors include a lot
of information that does not belong in an Introduction but should be placed in the study
site description (section 2) or in a classic Methods section. The last two paragraphs need
to be thoroughly reorganized and shortened. They could be combined with the well-
defined aim of the study (P3-L11-14).

P1-L26:      I don't think "duration" (of the database ) is the right term here; consider
             using "ii) a relatively short temporal coverage;"

P1-L28:      Add a reference to underline this statement

P2-L1-5:     This is a very long sentence with a lot of parentheses. Consider simplifying
             the text by starting a new sentence with:
             "They cause deaths and destruction (buildings, tourism infrastructure, power
             lines, forest stands), sever communications…"

P2-L10:      The information in footnote 1 is too detailed to remain in the main text. If
             the authors want to keep it, they should consider including it in a list of
             additional information in a supplement.

P2-L20:      This is confusingly written, consider rephrasing, e.g.:
             "with only a few studies on this topic, such as Granet-Abisset…"

P3-L8/9:     The parenthesis is very long and complicated. Consider changing it to an
             own sentence.

P3-L10:      I do not understand what is meant by "a long lasting occupation of the
             territory". Do the authors mean "a long-term challenge for the public
             authorities"?

P3-L15-18: All the information describing the Vosges does not belong here; it should be incorporated in section 2. Part of it is actually already there (!).

P3-L19: The data in footnote 2 should probably be included in section 2 or handled in the same way suggested for footnote 1.

P3-L24/25: The description of the different types of sources should be presented in the Methods section, not here.

P3-L26-28: Dito. All this information on the social enquiries and surveys do not belong here (incorporate it in the Methods section). They only make the final part of the Introduction complicated

**Description of the territorial context**

This section is adequately structured but slightly long. I suggest that the second paragraph (P4-L17 to P5-L5) should be considerably shortened. A part of the information given there is interesting indeed, but not essential for the present study.

I like Figure 1 very much; it gives a good overview over the study region and nicely illustrates its main features.

P6-L1: Incorporate the content of the footnote in the text, if you think necessary.

**Geohistorical methodology**

Figure 2 gives an excellent overview over the sources that were evaluated in the study. It completes this section well.

P6-L9: Consider rephrasing to: "Because of the scarcity of data in the archives,"

P6-21/22: Simplify to: "…focused on the risk cultures of the practitioners of winter activities."

P6-L26/27: This is formulated in a complicated way. Maybe you could simplify: "This proportion is slightly below one-fifth for oral testimonies and regional media reports each."

P7-L5: At line 1 the authors use the term "ground observations"; here instead they use "direct field investigations and observations". Do they mean the same? I suggest rephrasing: "Direct field investigations including surveys of avalanche deposits"

**Results**

The Results section presents the most important findings of this contribution in two tables and three figures. It is of adequate length but quite confusingly written at times. The tables need a little bit of work (see comments below, in the according sub-chapter of

this review) and Figure 5 would greatly profit from the incorporation of cumulative data (see below, too).

Some statements on data uncertainty and accuracy do actually not belong in a classic Results section and should therefore be moved to the Discussion.

| | |
|---|---|
| P9-L19: | As suggested below (sub-chapter "Tables" of this review) the authors need to clarify how the term "casualties" is used in their MS. My impression tells me they mean "affected people", but I am not sure. |
| P9-L20: | How is "material damage" defined in this study? Does it only cover damage to buildings or is it also including damage to infrastructure (e.g. roads, train tracks, power lines etc.). Have the authors considered to use the term "financial damage" instead? |
| P9-L21/22: | See above right above: how is "functional damage" exactly defined? Does it have a financial aspect? Or put differently, does a functional damage to "road cuttings" include the damaging or destruction of infrastructure or only the blocking of a transportation route for some time? Please clarify these points |
| P9-L23-26: | These three sentences describe and discuss data limitation and data uncertainty. They do not really belong here and I recommend that they are incorporated in section 5 (Discussion) |
| P9-L26: | The information given in footnote 5 should actually be introduced in the Methods section. Alternatively it could be placed in the text here in section 4. |
| P10-L2-6: | These two sentences discuss data accuracy and do actually not belong in this section. I recommend that the point made here is included in section 5 (Discussion) |
| P10-L12: | In this paragraph (lines7 to 13), you round all the values of Table 3 with the exception of the "441 m a.s.l." at line 12. Please be consistent in the way you describe your data. |
| P11-L14/15: | Be exact: not the number occurs, the event/avalanche does. Hence, change to: "(about 5% of all avalanches of the chronology occurred…" |
| P11-L15: | I suggest you change to "The number of avalanches per winter rise in the 1960s, and increase even more… |
| P13-L1/2: | When stating the "negative correlation between avalanches intensity and cold season", do you mean that avalanche intensity shows a statistically significant negative trend with time? |
| P13-L2: | Footnote 6: I recommend stating in the text which method/test the authors applied. However, the second part of the footnote is not necessary |
| P13-L3-5: | Regarding the "relative proportion of the types of avalanches over time": I do not see the statement you make in this sentence in Figure 6, unless you are only talking about material damage. Please be a bit more concise. |

P13-L9/10:     What exactly is a "victim" in your context? A person that was injured or killed? Please explain. Also consider my comments regarding your definition of "casualties", e.g. in Table 2.
When you talk of "unscathed people" at line 10, I see a problem. Because in my opinion, somebody that was not killed or hurt during an event is not a "victim" in the proper sense. You could possibly call this "unscathed" person *affected* by the avalanche.

P13-L7:        Footnote 7: This doesn't work like this: The information clearly belongs in the Discussion section and not here in the Results section.

P13-L11:       After "material", "functional" and "environmental" damage, the authors introduce "human damage". What is exactly meant? Fatalities or both injured and killed people? Please explain. However, I recommend avoiding this term in the MS.

P14-L6-8:      First, I am not sure if listing all the years in the text helps the reader. Please reconsider that.
Then, the authors have listed the winter 2009/2010 twice (both in the winters with considerable victims and damage and in the slightly less affected winters). Please chose one of the two.

P15-L1:        Please note that February 1844 falls into the winter of 1843/44 which is not listed at the bottom of page 14 (1844/45 is listed instead)

P15-L10:       What is meant by "spatial extension of the information"? Extension of the affected area?

**Discussion**

Some parts of this section are not formulated well enough, which makes it hard for the reader to follow the central theme in the text. Also, some statements made in the Discussion are repetitive. Below I only list the points that I think are most important.

P15-L22/23:    I do not understand the definition of the first factor considered by the authors. Please try to explain this more clearly. Especially the second part of the statement ("leading or not to an 'event building' from the facts, and to their transmission") is confusing to me

P15-L27:       This title is a bit puzzling; I suggest something clearer

P16-L1/2:      This statement is not well reflected in Figure 5 because only count data are shown in the two graphs. Consider adding the cumulative number of events and sources

P16-L25:       "The changes to the body of available (re)sources are visible in the shape of the geo-chronology."
Where is this visible? In Figure 5? Please add reference to the figure after this sentence or after the next sentence.

P16-L28:       What do the authors mean by "consecutive"? I do not understand. Consider "…more frequently ever since recreational activities are undertaken"

| P16-L33/34: | What do the authors mean in the second part of the sentence "…with some years with very few avalanches."? Does the number of sources increase linearly with the number of avalanches? Have the authors made that plot (y-axis: sources per year; x-axis: avalanches per year)? |
|---|---|
| P17-L4: | The formulation "the question of the existence of potential occurrences" seems quite difficult to understand; if possible simplify/clarify |
| P18-L2/3: | This sentence ("The observation… …winter sports") is true but not of great importance for the article and should thus be deleted to shorten the text |
| P18-L6: | Consider deleting "and even more from 1993 to 1994," it makes the sentence even harder to read |
| P18-L11-23: | This paragraph is confusing me. I wonder if a clear and well written definition of terms used in the MS could help and clarify some points. Such a list of definitions would probably have to be placed in section 3 and would include terms like: trace, occurred avalanche, event (source event, historical event, observed events, avalanche event), event building, source, source effect etc. The complicated footnote 10 includes such a definition ("…we see as 'event' all spatio-temporal occurrences of the avalanche phenomenon."). However, there it is not helping very much. |
| P19-L7/8: | I suggest to clarify slightly as follows: "…no mention is made of smaller avalanches (intensity class < 3). In contrast,…" |
| P19-L8: | is "damage/size levels" referring to intensity classes? Please be consistent in the use of these terms (also see my comment regarding the legend of Figure 8). |
| P20-L1/2: | This first sentence is a very good example of a slightly complicated sentence that could be simplified, e.g. as follows: "Land use in the Vosges Massif has significantly changed over the 240.year study period." |
| P21-L2-43: | This sentence is repetitive |
| P21-10/11: | I am not sure if these questions raised by the authors help here. I would definitely delete the third one to help shorten the text |
| P21-L14: | Footnote 15 can compactly be integrated into the text here; consider: "… rather than written culture. This was also suggested for other mountainous areas (Barrué-Pastor (2014)." |
| P21-L17-20: | Consider simplifying: "Moreover, it is not sure if avalanches were really perceived as a significant threat since the chronology does not include many avalanches that caused material damage or fatalities." |

| P22-L7/8: | The sentence "Finally, it implies that the memory of the risk is in the 'short time of the event, experienced during a lifetime' (Barrué-Pastor, 2014)." is confusing |
| P22L28-P23-L1: | Very long sentence; please consider making two sentences |

**Conclusions**

This section is of adequate length and contains the important take home messages for the reader. However, it needs to be sharpened here and there.

| P23-L10-13: | This sentence is way too long and difficult. I recommend dividing the information in two or more clear sentences. |
| P24-L1-5: | Again, this sentence is way too long. Start new sentence at line 4 and consider changing to, e.g.: "This applies particularly to medium high regions where knowledge is still very partial." |
| P24-L6/7: | Again, I suggest clarifying the text by deleting ", in fact," at line 6 and by starting a new sentence at line 7:
"… both in terms of frequency and intensity. Avalanches caused a dozen deaths since 1970, which makes it one of the deadliest natural hazards in the Alsace region". |
| P24-L8-10: | Complicated confusing sentence again. Simplify |
| P24-L15/16: | Consider deleting the first part of the sentence and changing to:
"The geo-chronology reflects only part of the avalanches that actually occurred during the study period." |
| P24-L18: | What is meant by "the prism of the corpus of sources"? |
| P24-L21-23: | Consider simplifying:
"…at least in some sectors. These factors lead to a reduction in event frequency and intensity, and even to the disappearance of avalanches in certain valley sites. |
| P24L30-P25L2: | I would have wished a slightly more down-to-earth last paragraph. Although I fully agree that available sources strongly influence avalanche occurrence as we perceive it (message of the first sentence), I find that especially the second but also the third sentences are a bit confusing. |

**Tables**

| Table 2: | A number (28) is missing for "Large avalanches (intensity class > 3)" |
| Table 2: | In the caption of table 2 the authors should define how they use the term "casualties". If here "casualties" means "affected people", I would strongly suggest to use "affected people" (i.e. concerned by an avalanche but neither injured nor killed). |

| Table 2: | I suggest to use the term "With fatalities" or "With people killed" instead of "With dead people" |
| --- | --- |
| Table 2: | The different types of damage (material, functional and environmental) should in my opinion be accurately defined/explained in the caption of Table 2. Also, I suggest to use the "damage" in singular everywhere in Table 2. |
| Table 3: | In the caption, consider changing text to "Topographic characteristics of the avalanche paths…" |
| Table 3: | It seems strange to me to have listed the mean altitude of the avalanche paths. How was it calculated? I guess the table would considerably benefit from information on the (i) starting zone and (ii) the bottom of the depositional area of the different paths |
| Table 3: | The unit of "Mean altitude" should be "m a.s.l." |

**Figures**

| Figure 2: | (a) Consider slightly rephrasing the figure caption: "Overview over the set of geohistorical resources used in the present study" |
| --- | --- |
| Figure 2: | In the upper-left-corner "Written re(sources)" should be changed to "Written (re)sources". |
| Figure 2: | The fourth box of "Written (re)sources" needs to be translated in English. Also, in the third box "Regional" has no *accent aigu* in English |
| Figure 4: | In the caption, consider changing text to:
"Only the 520 avalanches that could be precisely located are considered." |
| Figure 5: | The data presented in the two graphs are interesting. However, they will be better shown additionally using a cumulative chart of both avalanche events (A) and sources (B). I strongly encourage to add cumulative data to both panels |
| Figure 6: | As blue is the only color used in this figure, I suggest to only place the text in the seven panels and leave the colored rectangle out |
| Figure 6: | Please consider my comments made above (Table 2) regarding affected/injured/killed people and regarding the damage types |
| Figure 7: | I suggest you chose an alternative indication for the two WWs; the dashed line should only be used to show the sub-periods |
| Figure 7: | What is the difference between the green (Data contextualization) and the red (Data and data contextualization) symbols? Briefly explain in the figure caption |
| Figure 7: | In the legend, use "sub-period" instead of "period" |

Figure 8:    In the caption, consider changing to:
             "Location and intensity of the avalanches that were reported in the
             Honneck-Rothenbachkopf sector during the winters 1951/52 (left) and
             2009/10 (right), obtained from the geochronology developed in this
             study."

Figure 8:    In the legend of Figure 8 the authors use the term "Level" instead of
             "intensity class" (see also Table 2). I strongly recommend to use the
             same term throughout the text

**3) Selection of technical corrections**

| | |
|---|---|
| P1-L25: | You need a semicolon or colon after "i) a close link to vulnerability". A comma doesn't work here. |
| P1-L28: | Why "the period"? Which period is meant here? |
| P1-L25: | It should probably read "result" instead of "results" |
| P2-L12: | Change to: "the latter region has had" |
| P2-L16: | Consider "probability instead of "chance" |
| P2-L22: | Consider starting a new sentence at line 23: "However, the study of the evolution…" |
| P3-L7: | I guess you need "and" before "vulnerability" |
| P3-L11: | Be consistent in spelling "geohistorical" ("geo-historical" at page 9, line 17) |
| P3-L12: | Consider changing to "approach for the" |
| P3-L12: | Consider "a geo-chronology of avalanches and avalanche damage of" |
| P3-L27: | Be consistent in spelling "Vosges Massif" throughout the text. Also, the use of "Vosges Range"/"Vosges range" is nor consistent! |
| P4-L4 | Consider changing "west" to "western" |
| P4-L7 | I suggest the use of abbreviations: "km$^2$" instead of "square kilometers", "m a.s.l." instead of "meters" etc. |
| P4-L7/8: | Here and in many other places in the text, the authors spelled out numbers instead of using figures/numerals. Personally I think that in many cases, the use of figures would be more appropriate (in this case "between 20 and 60 km" instead of "twenty and sixty km"). Again, consistency is the key. |
| P4-L13: | Consider deleting "long-lasting" (you state right afterwards that it persists until spring or even later). |
| P4-L16: | Reformulate: "is 20%, 30% and 60% at 700, 1000 and 1350 m a.s.l., respectively (Wahl et al., 2009)." |
| P5-L3: | Consider adding commas: "This, in turn, had a…" |
| P6-L1: | I think that the Vosges are plural in English, too. Hence, change "has never been" to "have never been" |
| P6-L5: | Consider changing to: "was partitioned at its main ridge into two separate" |
| P6-L6: | Maybe use "official language" instead of "language" |

| | |
|---|---|
| P6-L10: | Consider using "data set" instead of "corpus" |
| P6-L16/17: | Simplify to "increased human activity in the" |
| P6-L19: | Change to "contributions, ranging from" |
| P6-L27: | Consider using "118" in figures. |
| P7-L4: | Change to "areas where past" |
| P7-L7: | Consider rephrasing: "All these (re)sources were sufficiently abundant…" |
| P7-L12/13: | Confusing, consider changing to:
"However, it was not always possible to…" |
| P9-L24: | Delete the second "only" at the end of the sentence (repetitive) |
| P9-L27: | Delete "therefore", it is not necessary |
| P10-L1-2: | Consider slightly changing to: "Over the whole period, more than 90% of all the 730 identified avalanches (682 events) could be specifically related…" |
| P10-L7: | Consider changing to "Locating the path of an avalanche was possible for 520 of the 730 events." |
| P10-L7-9: | A bit awkwardly put; consider:
"All others, except one, have been associated with a sector (geographical area of a few square kilometers in size in which several avalanche paths are located." |
| P10-L9: | Change to
"the characteristics of typical avalanche paths (Table 3). Its length …" |
| P10-L10: | Here and below, use abbreviations "m" or "m a.s.l." for "meters" |
| P10-L11: | Change "forty meters" to "just under 40 m" |
| P11-L5: | Change to "… in the southern part of the massif in the High Vosges mountains (more than 95% of all events)." |
| P11-L6/7: | Consider changing to: "… are mainly oriented to the north east, east and the south east and, to a lesser extent, to the south." |
| P11-L8: | Consider changing to "…of cornices and subsequent avalanches mentioned above." |
| P12-L4: | Change to "almost always high intensity events (greater than 3 on the scale previously introduced; Table 2)." |
| P12-L6: | Change to "…difference between the pre-1990 records and the more complete recent records is that the number of recorded low intensity events (intensity less than or equal to three) drastically increased in the latter period (Fig. 6). |

| | |
|---|---|
| P13-L10: | Change to "…half of these have been" |
| P15-L6: | change "cubic meters" to "m$^3$" |
| P15-L11/12: | Consider changing to:
"…apart from a few exceptions, avalanches did not occur in the valleys anymore in this most recent period. |
| P15-L28: | I am not sure what the authors mean by "net" (clear, considerable?). |
| P16-L1/2: | I guess the authors mean "between the period from the 1940s to the early 1990s and the period covering the 1990s to today" |
| P16-L5: | I suggest the use of "sub-periods" instead of "periods" |
| P16-L5/6: | Consider changing to:
"…based on the predominance of different types of (re)sources available during the entire study period (Fig. 7)." |
| P16-L22: | Change "exponential" to "considerable" |
| P16-L25: | Consider starting the sentence with "The" instead of "These" |
| P16-L27: | Instead of "…from when the registering of avalanches becomes more regular, a change linked…" consider using "…when avalanches were more regularly registered. This change is linked…" |
| P16-L31: | Change "this data" to "then" and change "a winter" to "per winter" |
| P17-L9: | Consider using "since" instead of "from" |
| P17-L10: | Instead of "whatever", consider "regardless of" |
| P19-L8: | Consider changing to "many small intensity events" |
| P21-L1: | Consider "The increase of visits partly explains…" |
| P21-L9: | Consider adding a reference: "…mid-twentieth century (Fig. 5B)…" |
| P22-L9: | Change to "…from persona, experiences…" |
| P22-L11: | Consider changing to: "this study was carried out in a" or "this study is associated to a" |
| P22-L12: | Delete "already" |
| P22-L13: | Instead of the "quality of the snow," I recommend to write "the characteristics of the snowpack," or "the structure of the snowpack," |
| P22-L19: | Consider deleting "global" |
| P22-L26: | Delete "visible" |
| P23-L6: | Consider changing to: "The absence or scarcity of snow is…" |
| P23-L21: | Delete comma after "the analysis exploits" |

P24-L26:           Use a full stop instead of an exclamation mark

P25-L1:             Change to "The reproduction of this type of…"

---

## Referee Comment (RC2) · Anonymous Referee #2 · 21 Feb 2017

The paper presented by Giacona et al. deals with an interesting topic and even more addresses the approach to a region which has not been know so far for its avalanche activity. As such, the reconstruction presented is of high relevance, as these low-elevation mountain ranges, as the Vosges, are likely to be the first and most severely affected by climate change and could thus serve as examples/illustrations of what one needs to expect at higher altitudes. Also, the number of sources collected by the authors is impressive and shows the great ability of the team to cross natural and human science approaches, which is still rare.

The main weaknesses of the paper reside in the way it was written, and this for several reasons. First of all, and despite the fact that the authors acknowledge a native speaker for the proof-reading, the text is written in a rather poor English, and many technical words have been translated simply from French (such that they either have a different

or no more meaning in English). The language will need to be polished substantially in a new version.

The style of the paper also seems awkward to a natural scientist as it uses footnotes (and even in large numbers), which is all but normal in natural sciences (by contrast to human sciences). This needs to be clarified as well.

Thirdly, the manuscript is fairly descriptive in the introduction and not very clear either in the abstract. Overall, the text needs to become much more concise and focused, and also much clearer in view of terminology. What is a nonconventional source (for me, natural or written archives are conventional indeed, but not sufficiently used)? Where do you address hazards, and where are you really addressing risks? This is used in a mixed way and needs clarification as well. What are risk historians? What is a geohistorical methodology/resources/approach? etc. (I could provide many more examples, but would like to suggest that the authors stick to the international literature when using definitions or terms.

The results will need to be presented in a much clearer, and more organized way. There is more in the data than you are showing so far. In the same line of thoughts, please make sure that you put your data into a larger context, the discussion is very much focused at the case-study site so far and introduces many new results rather than seeing them in a broader context.

In my opinion the paper can become a very nice and relevant piece, and certainly will be suitable for NHESS, but more work is needed to reach this goal, and I would be happy to see a new version on that interesting topic sometimes soon.

---

## Author Comment (AC2) · 3 Apr 2017

Dear Referee,

We are very grateful for your encouragements and your valuable comments regarding the interest of our work. Special thanks for the very detailed editing work you have done. We agree that the previous version of the paper had several formal weaknesses that made its full benefit difficult to grasp, especially for people not familiar with a historical approach to natural hazards. We have addressed all the concerns raised as outlined in the detailed response attached to this comment. We would be glad to submit the revised version of the paper to NHESS. We sincerely hope that the changes made will make the article adequate for publication.

With best regards,

Florie Giacona, for the authors.

Please also note the supplement to this comment:
http://www.nat-hazards-earth-syst-sci-discuss.net/nhess-2016-395/nhess-2016-395-AC2-supplement.pdf

---

## Author Response (AR1)

**Cover letter for the revised version of our article: *"A 240 year History of Avalanche Risk in the Vosges Mountains from Nonconventional Sources"*, submitted to Natural Hazards and Earth System Sciences**

Dear Natural Hazards and Earth System Sciences editorial board, dear Dr. Sven Fuchs, dear referees,

Thank you very much for the publication of our article in Natural Hazards and Earth System Sciences Discussion, and for the encouragements and valuable comments conveyed by the two referee reports. These have been very useful for us to improve the paper. We have addressed each of their concerns as outlined below. The corresponding changes have been included in a revised version of the paper we have prepared and that we would be glad to submit to your kind consideration.

Both referee reports agreed upon the high interest of the topic of the paper, and upon the adequacy of the approach and methods we have employed. Also, the referees both agreed upon the need of solving several formal issues. According to this statement and to the detailed comments kindly provided by the referees, we have deeply reworked all the formal aspects of the paper. Modifications made include:
- A slight overall reorganization to better distinguish the content of the different sections. Also, the discussion section has been slightly expanded to put our results in perspective with regards to a larger context;
- A real effort for precising and defining the exact meaning of all the terms we are using, especially of those related to the typology of avalanche events, and also of those arising from the fields of history and social science (and therefore arguably less familiar to natural hazards (geo)scientists). To this end, we have included an additional table in the text core, which contains the definition of such terms;
- Suppression of all footnotes, with inclusion of the relevant information within the text core;
- Yet, significant shortening of the text core;
- Correction of all typos, awkward sentences, etc., and further English smoothing by a professional English corrector.

However, we want to stress that the first author has her background in history, and that the main outcome of the paper, in addition to specific findings for the case study, is the contribution of this discipline to the better understanding of the evolution of natural hazards on the long range. This has two consequences for the paper:
- First, the data/result/discussion organization may remain a bit different from the one of a pure geoscience paper. Specifically, analyzing the source amount, quality and evolution through time clearly belongs, for the historian, to the results section and not only to the discussion section. It is even one of the most important points of the work to illustrate that the two aspects cannot be truly distinguished.
- Second, the text style remains truly a bit more literary than in standard geoscience articles.
As stated before, we have polished the paper with regards to the first submission, which should contribute to make it easier and more convenient to read for the NHESS readership. Yet, we don't want to completely get rid of these intrinsic specificities of the work, which also contribute, in our opinion, to its value.

By the way, we provide in specific comments a point-by-point answer to the two referee reports in the online discussion.

All in all, we feel that the new version of the paper is much clearer and more precise, and hope that this will make it adequate for publication in Natural Hazards and Earth System Sciences.

With best regards,

Florie Giacona, for the authors.

**Author's response to Anonymous referee 1**

**1) General comments**

Dear Editor, dear Authors,

This contribution covers a very important and interesting topic, namely the search, analysis and exploitation of conventional and nonconventional information sources to establish and/or extend databases of past natural hazard events. The article points out the many challenges associated with data acquisition over a long study period during which information sources evolved substantially. The text is generally well illustrated and the overall structure of the manuscript is adequate.

Data on natural hazard processes (especially when destructive) are used by a wide variety of organisations (e.g. scientific institutes, private environment companies, insurance companies, governments). Such data are a very useful instrument in providing basic information for better hazard and risk assessment as well as decision-making. It is therefore important to promote and support studies such as the one presented in this manuscript.

The case study presented by the authors focuses on a single hazard type (snow avalanches) and on regional issues (Massif des Vosges). It is in many ways special which partly makes it even more valuable. Most importantly, (i) it covers a very long time frame, (ii) it covers an area not especially known for the hazard process of snow avalanches and (iii) it treats an area that has experienced fundamental turbulences in the past (e.g. several armed conflicts in the last 150 years; changes in official language etc.). These points reveal methodological research problems that one may not encounter in similar investigations elsewhere.

Given the importance of natural hazard event analyses and data sets, I think this valuable contribution should be published and I hope that it will motivate researches in other regions and dealing with other hazard processes to follow this (obviously quite demanding) path.

However (!), I see several substantial (mainly formal and structural) problems that need to be solved before this text is ready for publication in NHESS.

☐ First of all, the use of English language is not very good and needs to be substantially improved. In many places, the constructed sentences are much too long and contain too much information that make them often confusing and difficult to understand. The authors should try to formulate short, reader-friendly and clear sentences. I tried to make suggestions/corrections where possible. However, because I am not a native-English speaker my proofreading is not at all complete. I am strongly convinced that the text would benefit from thorough editing by a native-English speaker.

☐ Footnotes are used throughout the manuscript. I would strongly recommend to avoid the use of such footnotes. It is not, to the best of my knowledge, acceptable in this journal. The contents of many of these footnotes is not essential for the understanding of the text. They could e.g. be summarised in a supplementary file associated to the article or could partly be deleted. The most important contents should be incorporated in the main text or in figure captions.

☐ In my opinion, the manuscript is currently too long. A lot of the issues addressed in the text are described in too much detail. In some parts (e.g. Discussion), the text is repetitive. I think it is crucial that the authors rework the manuscript, mainly sections 4 and 5, by carefully picking the main statements they want to make. According their decision, these issues and statements need to be accurately put to paper in simple but significant sentences (see also first point above).

☐ Some key terms used throughout the manuscript should probably be introduced and precisely be defined in the Methods section. For example, the use of the term *event* (also *source event*, *historical event*, *observed events*, *avalanche event* etc.) is not clear and confusing at times. At page18/line15 the authors state that "it is not because an

avalanche occurs that the event exists" and then produce a definition from literature. This is really complicated to understand and should (in my opinion) be clarified earlier in the text.

In summary, I do not think that this manuscript is ready for publication yet. I suggest the authors revise their text and solve the formal problems mentioned above. As regards content, the article is on a good level. However, the information the authors want to communicate in the latter sections of the manuscript should be reassessed, reorganized and if possible shortened. I do sincerely think that there is potential for an NHESS article in this manuscript. I thus recommend that the paper be accepted pending major revisions or that it be rejected with an invitation to re-submit when all formal aspects criticised are clarified. I provide below a list of partly detailed comments specific to the different sections of the article as well as to the tables and figures produced by the authors. Additionally, I prepared a list of technical corrections for the authors.

Authors' response: Thank you very much for this detailed and constructive review. According to it, we have deeply reworked all the formal aspects of the paper. Modifications made include:
- A slight overall reorganization to better distinguish the content of the different sections. Also, the discussion section has been slightly expanded to put the results in perspectives with regards to a larger context;
- A real effort for precising and defining the exact meaning of all the terms we are using, especially those related to the typology of avalanche events, and also those arising from the fields of history and social science (and therefore arguably less familiar to natural hazards (geo)scientists). To this end, we have included an additional table which contains the definition of such terms;
- Suppression of all footnotes, with inclusion of the relevant information within the text core;
- Yet, significant shortening of the text core;
- Correction of all typos, awkward sentences, etc., and further English smoothing by a professional English corrector.

However, we want to stress that the first author has her background in history, and that the main outcome of the paper, in addition to specific findings for the case study, is the contribution of this discipline to a better understanding of the evolution of natural hazards on the long range. This has two consequences for the paper:
- First, the data/result/discussion organization may remain a bit different from a pure geoscience paper. Specifically, analyzing the source amount, quality and evolution through time clearly belongs, for the historian, to the results section and not only to the discussion section. It is even one of the most important points of the approach to show that the two aspects cannot be distinguished;
- Second, the text style remains a bit more literary than in standard geoscience articles.

As stated before, we have polished the paper with regards to the first submission, which should contribute to make it easier and more convenient to read for the NHESS readership. Yet, we don't want to completely get rid of these intrinsic specificities of the work, which also contribute, in our opinion, to its value. The additional table in the revised version of the paper should help making the bridge between historical / social science concepts and the field of natural hazards.

**2) Specific comments regarding the different sections of the article (Anonymous Referee #1)**

**Abstract**
The Abstract is well structured and of adequate length.
Authors' response: Thank you.

P1-L9/10: Why the use of "still"? Consider changing to: "especially in medium-high mountain ranges with a significant process activity."

Authors' response: OK, this has been done.

P1-L11: A real problem of this MS is the lack of consistency in the spelling of terms. A good example is the term "north-east of France" at line 11 that throughout the text is also spelled "North-East of France" and "North East of France".

P1-L12: Same point: "geo-chronology" is also spelled "geochronology" in the MS. Be consistent.

Authors' response: Thank you very much for this remark. We have made an additional effort through the whole text to keep the spelling consistent. We now use "geo-chronology" in the whole text, for example.

P1-L17/18: I do not understand what is meant by "for risk changes understanding and mitigation"

Authors' response: We have changed this to "for mitigating risk and understanding its change through time". This should be clearer.

**Introduction**

The Introduction is not so well organized. Towards the end of it the authors include a lot of information that does not belong in an Introduction but should be placed in the study site description (section 2) or in a classic Methods section. The last two paragraphs need to be thoroughly reorganized and shortened. They could be combined with the well-defined aim of the study (P3-L11-14).

Authors' response: OK, the paper has been reorganized as suggested.

P1-L26: I don't think "duration" (of the database) is the right term here; consider using "ii) a relatively short temporal coverage;"

Authors' response: OK, this has been done.

P1-L28: Add a reference to underline this statement

Authors' response: We have added reference to:

Von Kotze, A., Holloway, A. (1996). Reducing Risk: Participatory teaming activities for disaster mitigation in Southern Africa. IFRC - International Federation of Red Cross/ Red Cresent Societies report, 339p

P2-L1-5: This is a very long sentence with a lot of parentheses. Consider simplifying the text by starting a new sentence with: "They cause deaths and destruction (buildings, tourism infrastructure, power lines, forest stands), sever communications…"

Authors' response: OK, this has been done.

P2-L10: The information in footnote 1 is too detailed to remain in the main text. If the authors want to keep it, they should consider including it in a list of additional information in a supplement.

Authors' response: OK, this has been done.

P2-L20: This is confusingly written, consider rephrasing, e.g.: "with only a few studies on this topic, such as Granet-Abisset…"

Authors' response: OK, this has been done.

P3-L8/9: The parenthesis is very long and complicated. Consider changing it to an own sentence.

Authors' response: OK, this has been done.

P3-L10: I do not understand what is meant by "a long lasting occupation of the territory". Do the authors mean "a long-term challenge for the public authorities"?

Authors' response: No, it simply means "occupied since a long time". We have reformulated the sentence.

P3-L15-18: All the information describing the Vosges does not belong here; it should be incorporated in section 2. Part of it is actually already there (!).

Authors' response: OK, this has been done.

P3-L19: The data in footnote 2 should probably be included in section 2 or handled in the same way suggested for footnote 1.
Authors' response: OK, this has been done.
P3-L24/25: The description of the different types of sources should be presented in the Methods section, not here.
Authors' response: OK, this has been done.
P3-L26-28: Dito. All this information on the social enquiries and surveys do not belong here (incorporate it in the Methods section). They only make the final part of the Introduction complicated
Authors' response: OK, this has been done.

**Description of the territorial context**
This section is adequately structured but slightly long. I suggest that the second paragraph (P4-L17 to P5-L5) should be considerably shortened. A part of the information given there is interesting indeed, but not essential for the present study.
Authors' response: OK, this has been done.
I like Figure 1 very much; it gives a good overview over the study region and nicely illustrates its main features.
Authors' response: Thank you.
P6-L1: Incorporate the content of the footnote in the text, if you think necessary.
Authors' response: OK, this has been done. In addition, some elements that were previously in introduction have been incorporated to this section.

**Geohistorical methodology**
Figure 2 gives an excellent overview over the sources that were evaluated in the study. It completes this section well.
Authors' response: Thank you.
P6-L9: Consider rephrasing to: "Because of the scarcity of data in the archives,"
P6-21/22: Simplify to: "…focused on the risk cultures of the practitioners of winter activities."
Authors' response: OK, this has been done.
P6-L26/27: This is formulated in a complicated way. Maybe you could simplify: "This proportion is slightly below one-fifth for oral testimonies and regional media reports each."
Authors' response: OK, this has been done.
P7-L5: At line 1 the authors use the term "ground observations"; here instead they use "direct field investigations and observations". Do they mean the same? I suggest rephrasing: "Direct field investigations including surveys of avalanche deposits"
Authors' response: We agree, this has been changed as suggested.

Authors' comment: Regarding this section, please consider the additional table regarding the terms definition. We have also added precisions regarding the specific avalanche intensity class we are using, and how we define the different types of damages and avalanche accidents involving people. Finally, some elements that were previously in introduction have been incorporated to this section.

**Results**
The Results section presents the most important findings of this contribution in two tables and three figures. It is of adequate length but quite confusingly written at times. The tables need a little bit of work (see comments below, in the according sub-chapter of this review) and Figure 5 would greatly profit from the incorporation of cumulative data

(see below, too). Some statements on data uncertainty and accuracy do actually not belong in a classic Results section and should therefore be moved to the Discussion.

Authors' response: Thank you for this comment. We have rather deeply reworked the text of this section for more clarity. We have also added subsequent precisions in the method section and in corresponding table/figure captions. Finally, we have included the cumulative plot for events, which indeed nicely illustrate their temporal evolution. However, we did not add the same plot for sources, as we think that adding sources over successive years does not have that much sense.

Finally, some points have been moved to other sections. However, analyzing the source amount, quality and evolution through time clearly belongs, for the historian, to the results section and not only to the discussion section (see our general comment at the beginning of our answer), so that we kept some statements about data accuracy here. Also, we even moved the evolution of sources through time previously in the discussing section to this section, now divided in two sub-sections. This should lighten the discussion section and better highlight the methodological outcomes of the paper.

P9-L19: As suggested below (sub-chapter "Tables" of this review) the authors need to clarify how the term "casualties" is used in their MS. My impression tells me they mean "affected people", but I am not sure.

Authors' response: We have added in the method section and in the table caption "By casualties, we mean either with people killed, injured and/or simply caught by the flow without any harmful consequences."

P9-L20: How. is "material damage" defined in this study? Does it only cover damage to buildings or is it also including damage to infrastructure (e.g. roads, train tracks, power lines etc.). Have the authors considered to use the term "financial damage" instead?

P9-L21/22: See above right above: how is "functional damage" exactly defined? Does it have a financial aspect? Or put differently, does a functional damage to "road cuttings" include the damaging or destruction of infrastructure or only the blocking of a transportation route for some time? Please clarify these points

Authors' response: We have added in the method section: "Property damages refer mostly to partial or total destruction of one or several buildings. In rare cases, it also corresponds to damage to bridges or to fences. Functional damage refers to traffic road perturbation. Environmental damage refers to perturbation of ecosystems, up to the destruction of large forest stands." Note that we use property damages instead of material damages.

P9-L23-26: These three sentences describe and discuss data limitation and data uncertainty. They do not really belong here and I recommend that they are incorporated in section 5 (Discussion)

Authors' response: See our first answer to this section comments.

P9-L26: The information given in footnote 5 should actually be introduced in the Methods section. Alternatively it could be placed in the text here in section 4.

Authors' response: OK, this has been changed.

P10-L2-6: These two sentences discuss data accuracy and do actually not belong in this section. I recommend that the point made here is included in section 5 (Discussion)

Authors' response: See our first answer to this section comments.

P10-L12: In this paragraph (lines7 to 13), you round all the values of Table 3 with the exception of the "441 m a.s.l." at line 12. Please be consistent in the way you describe your data.

Authors' response: OK, this has been changed.

P11-L14/15: Be exact: not the number occurs, the event/avalanche does. Hence, change to: "(about 5% of all avalanches of the chronology occurred…"

Authors' response: OK, this has been changed.

P11-L15: I suggest you change to "The number of avalanches per winter rise in the 1960s, and increase even more…

Authors' response: OK, this has been changed.

P13-L1/2: When stating the "negative correlation between avalanches intensity and cold season", do you mean that avalanche intensity shows a statistically significant negative trend with time?

Authors' response: Yes, this has been changed.

P13-L2: Footnote 6: I recommend stating in the text which method/test the authors applied. However, the second part of the footnote is not necessary

Authors' response: OK, this has been changed and simplified.

P13-L3-5: Regarding the "relative proportion of the types of avalanches over time": I do not see the statement you make in this sentence in Figure 6, unless you are only talking about material damage. Please be a bit more concise.

Authors' response: We are a bit unsure about the meaning of this comment. However, we reformulated the idea which should be clearer now.

P13-L9/10: What exactly is a "victim" in your context? A person that was injured or killed? Please explain. Also consider my comments regarding your definition of "casualties", e.g. in Table 2. When you talk of "unscathed people" at line 10, I see a problem. Because in my opinion, somebody that was not killed or hurt during an event is not a "victim" in the proper sense. You could possibly call this "unscathed" person *affected* by the avalanche.

Authors' response: As said before, we have added in the method section and in the table caption: "By casualties, we mean either with people killed, injured and/or simply cached by the flow without any harmful consequences." We now stick in the whole text on this specific typology.

P13-L7: Footnote 7: This doesn't work like this: The information clearly belongs in the Discussion section and not here in the Results section.

Authors' response: OK, this has been moved to the new sub-section putting our results in perspective with regards to other contexts.

P13-L11: After "material", "functional" and "environmental" damage, the authors introduce "human damage". What is exactly meant? Fatalities or both injured and killed people? Please explain. However, I recommend avoiding this term in the MS.

Authors' response: We agree that this was a weird way of saying casualties. We not stick on the term "with casualties" for all avalanches with physically affected people (i.e., either with people killed, injured and/or simply caught by the flow without any harmful consequences).

P14-L6-8: First, I am not sure if listing all the years in the text helps the reader. Please reconsider that. Then, the authors have listed the winter 2009/2010 twice (both in the winters with considerable victims and damage and in the slightly less affected winters). Please chose one of the two.

Authors' response: We agree that the exhaustive list of winters does not bring much to the analysis and simply removed it.

P15-L1: Please note that February 1844 falls into the winter of 1843/44 which is not listed at the bottom of page 14 (1844/45 is listed instead)

Authors' response: OK, but note that the sentence has been removed (see previous comment).

P15-L10: What is meant by "spatial extension of the information"? Extension of the affected area?

Authors' response: No, simply the spatial distribution of avalanche observations. This has been changed.

**Discussion**

Some parts of this section are not formulated well enough, which makes it hard for the reader to follow the central theme in the text. Also, some statements made in the Discussion are repetitive. Below I only list the points that I think are most important.

Authors' response: Thank you for this comment. We agree that the text of this section was hard to follow at some points and sometimes a bit long-winded. We therefore deeply reworked and restructured it, in order to smooth and polish the text (see our general comment), and to avoid repetitions as much as possible. We have also moved some points to the result section (the sources temporal evolution) and put added one new sub-section to put our results in perspective with regards to other contexts.

P15-L22/23: I do not understand the definition of the first factor considered by the authors. Please try to explain this more clearly. Especially the second part of the statement ("leading or not to an 'event building' from the facts, and to their transmission") is confuse other to me.

Authors' response: This short paragraph has been largely reworked and shortened. The idea is now to just introduce the discussion about the origin of the geo-chronology without using terms that have not been defined before. We come back on this in conclusion when we summarize our main findings.

P15-L27: This title is a bit puzzling; I suggest something clearer

Authors' response: We agree that this terminology in not standard. Yet, "source effect" and "event building" are important concepts of the study, which are now defined in the new table (now Table 1), so that we keep our title, with the terms quoted and further discussed/explained in text.

P16-L1/2: This statement is not well reflected in Figure 5 because only count data are shown in the two graphs. Consider adding the cumulative number of events and sources

Authors' response: We have added the cumulative plots for events which indeed nicely illustrate their temporal evolution. However, we did not add the same plot for sources, as we think that adding sources over successive years does not have that much sense.

P16-L25: "The changes to the body of available (re)sources are visible in the shape of the geo-chronology." Where is this visible? In Figure 5? Please add reference to the figure after this sentence or after the next sentence.

Authors' response: Yes, in Figure 5 (now Figure 4). Reference to it has been inserted in text.

P16-L28: What do the authors mean by "consecutive"? I do not understand. Consider "…more frequently ever since recreational activities are undertaken"

Authors' response: OK, this has been done.

P16-L33/34: What do the authors mean in the second part of the sentence "…with some years with very few avalanches."?

Authors' response: Yes, this has been clarified.

Does the number of sources increase linearly with the number of avalanches? Have the authors made that plot (y-axis: sources per year; x-axis: avalanches per year)?

Authors' response: Yes, the linear relation between, the two quantities is rather strong (correlation coefficient of 0.93 between the number of avalanches and distinct sources, *p*-value <0.0001). This information has been inserted in text. As a consequence, the scatter plot, provided below, is for us not so necessary.

[Figure]

P17-L4: The formulation "the question of the existence of potential occurrences" seems quite difficult to understand; if possible simplify/clarify

Authors' response: OK, this has been reformulated.

P18-L2/3: This sentence ("The observation... ...winter sports") is true but not of great importance for the article and should thus be deleted to shorten the text

Authors' response: OK, it has been deleted.

P18-L6: Consider deleting "and even more from 1993 to 1994," it makes the sentence even harder to read

Authors' response: It is for us important information, so we kept it. However, we reformulated to be clearer.

P18-L11-23: This paragraph is confusing me. I wonder if a clear and well written definition of terms used in the MS could help and clarify some points. Such a list of definitions would probably have to be placed in section 3 and would include terms like: trace, occurred avalanche, event (source event, historical event, observed events, avalanche event), event building, source, source effect etc. The complicated footnote 10 includes such a definition ("...we see as 'event' all spatio-temporal occurrences of the avalanche phenomenon."). However, there it is not helping very much.

Authors' response: We agree. This has been very deeply reworked. All definition have been moved to Table 1 (the new table), and only the results discussion have been kept here.

P19-L7/8: I suggest to clarify slightly as follows: "...no mention is made of smaller avalanches (intensity class < 3). In contrast,..."

Authors' response: OK, this has been changed.

P19-L8: is "damage/size levels" referring to intensity classes? Please be consistent in the use of these terms (also see my comment regarding the legend of Figure 8).

Authors' response: Yes, we now speak of avalanche intensity level everywhere in the paper.

P20-L1/2: This first sentence is a very good example of a slightly complicated sentence that could be simplified, e.g. as follows: "Land use in the Vosges Massif has significantly changed over the 240.year study period."

Authors' response: OK, this has been changed.

P21-L2-43: This sentence is repetitive

Authors' response: OK, text has been reformulated and shortened.

P21-10/11: I am not sure if these questions raised by the authors help here. I would definitely delete the third one to help shorten the text

Authors' response: We agree, text here has been significantly shortened.

P21-L14: Footnote 15 can compactly be integrated into the text here; consider: "… rather than written culture. This was also suggested for other mountainous areas (Barrué-Pastor (2014)."

Authors' response: This has been moved to the discussion sub-section where the comparison with other contexts is proposed. It has also been slightly reformulated.

P21-L17-20: Consider simplifying: "Moreover, it is not sure if avalanches were really perceived as a significant threat since the chronology does not include many avalanches that caused material damage or fatalities."

Authors' response: OK, this has been done.

P22-L7/8: The sentence "Finally, it implies that the memory of the risk is in the 'short time of the event, experienced during a lifetime' (Barrué-Pastor, 2014)." is confusing

Authors' response: OK, it has been reformulated to be clearer.

P22L28-P23-L1: Very long sentence; please consider making two sentences

Authors' response: OK, this has been done.

**Conclusions**

This section is of adequate length and contains the important take home messages for the reader. However, it needs to be sharpened here and there.

Authors' response: Thank you for this comment. We have slightly reworked this section to take into account the different comments made, and also to better highlight two points raised by the referee two: i) better discussing the distinction between hazard and risk in the geo-chronology, which relates to the close link between events and vulnerability in the past, and, ii) the possibility to expand the exploitation of the avalanche chronology in the future, not that it is available and that a detailed contextualization has been done.

P23-L10-13: This sentence is way too long and difficult. I recommend dividing the information in two or more clear sentences.

Authors' response: OK, this has been changed.

P24-L1-5: Again, this sentence is way too long. Start new sentence at line 4 and consider changing to, e.g.: "This applies particularly to medium high regions where knowledge is still very partial."

Authors' response: OK, this has been changed.

P24-L6/7: Again, I suggest clarifying the text by deleting ", in fact," at line 6 and by starting a new sentence at line 7: "… both in terms of frequency and intensity. Avalanches caused a dozen deaths since 1970, which makes it one of the deadliest natural hazards in the Alsace region".

Authors' response: OK, this has been changed.

P24-L8-10: Complicated confusing sentence again. Simplify

Authors' response: OK, this has been changed.

P24-L15/16: Consider deleting the first part of the sentence and changing to: "The geo-chronology reflects only part of the avalanches that actually occurred during the study period."

Authors' response: OK, this has been changed.

P24-L18: What is meant by "the prism of the corpus of sources"?

Authors' response: This is a rather common social science expression suggesting that the source corpus gives a biased/specific vision of reality (here of avalanche activity and risk), as a prism transforms and decomposes the light. For clarity, we however reformulated the sentence.

P24-L21-23: Consider simplifying: "…at least in some sectors. These factors lead to a reduction in event frequency and intensity, and even to the disappearance of avalanches in certain valley sites.

Authors' response: OK, this has been changed.

P24L30-P25L2: I would have wished a slightly more down-to-earth last paragraph. Although I fully agree that available sources strongly influence avalanche occurrence as we perceive it (message of the first sentence), I find that especially the second but also the third sentences are a bit confusing.

Authors' response: Text has been clarified as much as possible. Also, according to our general comment for this section, new ideas have been incorporated.

**Tables**

Table 2: A number (28) is missing for "Large avalanches (intensity class > 3)"

Authors' response: thank you, this has been corrected. By the way, for consistency, we now use intensity level instead of intensity class.

Table 2: In the caption of table 2 the authors should define how they use the term "casualties". If here "casualties" means "affected people", I would strongly suggest to use "affected people" (i.e. concerned by an avalanche but neither injured nor killed).

Authors' response: We do not really like the term "affected people" since people can be affected by an avalanches even if they are not physically affected (psychological consequences, economic consequences due to delays, etc.). Therefore, we keep "casualties" but we have added in the method section and in the table caption "By casualties, we mean either with people killed, injured and/or simply caught by the flow without any harmful consequences."

Table 2: I suggest to use the term "With fatalities" or "With people killed" instead of "With dead people"

Authors' response: OK, we now use "with people killed".

Table 2: The different types of damage (material, functional and environmental) should in my opinion be accurately defined/explained in the caption of Table 2.

Authors' response: OK. We have added the definitions in the table caption, and more details are now provided in the method section.

Also, I suggest to use the "damage" in singular everywhere in Table 2.

Authors' response: OK, this has been changed.

Table 3: In the caption, consider changing text to "Topographic characteristics of the avalanche paths…"

Authors' response: OK, this all has been changed.

Table 3: It seems strange to me to have listed the mean altitude of the avalanche paths. How was it calculated? I guess the table would considerably benefit from information on the (i) starting zone and (ii) the bottom of the depositional area of the different paths.

Authors' response: Thank you. Such information was indeed lacking. We have added in the method section that all path profiles have been mapped in a GIS environment. We have also added information regarding the maximal and minimal altitudes of the paths profiles which corresponds to maximal altitude of the release area and minimal altitude of the runout zone, respectively. Finally we have precised in the table caption: "These [altitude statistics] were obtained by crossing the path GIS shape files with a 5m resolution Digital Elevation Model."

Table 3: The unit of "Mean altitude" should be "m a.s.l."

Authors' response: OK, this has been changed.

**Figures**

Figure 2: (a) Consider slightly rephrasing the figure caption: "Overview over the set of geohistorical resources used in the present study"

Authors' response: OK, this has been done.

Figure 2: In the upper-left-corner "Written re(sources)" should be changed to "Written (re)sources".

Authors' response: OK, this has been done.

Figure 2: The fourth box of "Written (re)sources" needs to be translated in English. Also, in the third box "Regional" has no *accent aigu* in English

Authors' response: OK, this has been done.

Figure 4: In the caption, consider changing text to: "Only the 520 avalanches that could be precisely located are considered."

Authors' response: OK, this has been done.

Figure 5: The data presented in the two graphs are interesting. However, they will be better shown additionally using a cumulative chart of both avalanche events (A) and sources (B). I strongly encourage to add cumulative data to both panels

Authors' response: We have added the cumulative plots for events which indeed nicely illustrate their temporal evolution. However we did not add the same plot for sources, as we think that adding sources over successive years does not have that much sense.

Figure 6: As blue is the only color used in this figure, I suggest to only place the text in the seven panels and leave the colored rectangle out

Authors' response: OK, this has been done.

Figure 6: Please consider my comments made above (Table 2) regarding affected/injured/killed people and regarding the damage types.

Authors' response: Please see our response above.

Figure 7: I suggest you chose an alternative indication for the two WWs; the dashed line should only be used to show the sub-periods

Authors' response: OK, this has been changed.

Figure 7: What is the difference between the green (Data contextualization) and the red (Data and data contextualization) symbols? Briefly explain in the figure caption

Authors' response: "Contextualization" means that some sources/information where used to describe the historical context only, whereas "Avalanche data and their contextualization" means that other sources were used to build the geo-chronology and understand the context of the corresponding events. The figure caption has been slightly changed and expanded to precise this.

Figure 7: In the legend, use "sub-period" instead of "period"

Authors' response: OK, this has been changed.

Figure 8: In the caption, consider changing to: "Location and intensity of the avalanches that were reported in the Honneck-Rothenbachkopf sector during the winters 1951/52 (left) and 2009/10 (right), obtained from the geochronology developed in this study."

Authors' response: OK, this has been changed.

Figure 8: In the legend of Figure 8 the authors use the term "Level" instead of "intensity class" (see also Table 2). I strongly recommend to use the same term throughout the text

Authors' response: OK. For consistency, we now use intensity level instead of intensity class everywhere in the paper.

**3) Selection of technical corrections**

P1-L25: You need a semicolon or colon after "i) a close link to vulnerability". A comma doesn't work here.

P1-L28: Why "the period"? Which period is meant here?

P1-L25: It should probably read "result" instead of "results"

P2-L12: Change to: "the latter region has had"

P2-L16: Consider "probability instead of "chance"

P2-L22: Consider starting a new sentence at line 23: "However, the study of the evolution…"

P3-L7: I guess you need "and" before "vulnerability"

P3-L11: Be consistent in spelling "geohistorical" ("geo-historical" at page 9, line 17)

P3-L12: Consider changing to "approach for the"

P3-L12: Consider "a geo-chronology of avalanches and avalanche damage of"

P3-L27: Be consistent in spelling "Vosges Massif" throughout the text. Also, the use of "Vosges Range"/"Vosges range" is nor consistent!

P4-L4 Consider changing "west" to "western"

P4-L7 I suggest the use of abbreviations: "km$_2$" instead of "square kilometers", "m a.s.l." instead of "meters" etc.

P4-L7/8: Here and in many other places in the text, the authors spelled out numbers instead of using figures/numerals. Personally I think that in many cases, the use of figures would be more appropriate (in this case "between 20 and 60 km" instead of "twenty and sixty km"). Again, consistency is the key.

P4-L13: Consider deleting "long-lasting" (you state right afterwards that it persists until spring or even later).

P4-L16: Reformulate: "is 20%, 30% and 60% at 700, 1000 and 1350 m a.s.l., respectively (Wahl et al., 2009)."

P5-L3: Consider adding commas: "This, in turn, had a…"

P6-L1: I think that the Vosges are plural in English, too. Hence, change "has never been" to "have never been"

P6-L5: Consider changing to: "was partitioned at its main ridge into two separate"

P6-L6: Maybe use "official language" instead of "language"

P6-L10: Consider using "data set" instead of "corpus"

P6-L16/17: Simplify to "increased human activity in the"

P6-L19: Change to "contributions, ranging from"

P6-L27: Consider using "118" in figures.

P7-L4: Change to "areas where past"

P7-L7: Consider rephrasing: "All these (re)sources were sufficiently abundant…"

P7-L12/13: Confusing, consider changing to: "However, it was not always possible to…"

P9-L24: Delete the second "only" at the end of the sentence (repetitive)

P9-L27: Delete "therefore", it is not necessary

P10-L1-2: Consider slightly changing to: "Over the whole period, more than 90% of all the 730 identified avalanches (682 events) could be specifically related…"

P10-L7: Consider changing to "Locating the path of an avalanche was possible for 520 of the 730 events."

P10-L7-9: A bit awkwardly put; consider: "All others, except one, have been associated with a sector (geographical area of a few square kilometers in size in which several avalanche paths are located."

P10-L9: Change to "the characteristics of typical avalanche paths (Table 3). Its length …"

P10-L10: Here and below, use abbreviations "m" or "m a.s.l." for "meters"

P10-L11: Change "forty meters" to "just under 40 m"

P11-L5: Change to "… in the southern part of the massif in the High Vosges mountains (more than 95% of all events)."

P11-L6/7: Consider changing to: "… are mainly oriented to the north east, east and the south east and, to a lesser extent, to the south."

P11-L8: Consider changing to "…of cornices and subsequent avalanches mentioned above."

P12-L4: Change to "almost always high intensity events (greater than 3 on the scale previously introduced; Table 2)."

P12-L6: Change to "…difference between the pre-1990 records and the more complete recent records is that the number of recorded low intensity events (intensity less than or equal to three) drastically increased in the latter period (Fig. 6).

P13-L10: Change to "…half of these have been"

P15-L6: change "cubic meters" to "m$_3$"

P15-L11/12: Consider changing to: "…apart from a few exceptions, avalanches did not occur in the valleys anymore in this most recent period.

P15-L28: I am not sure what the authors mean by "net" (clear, considerable?).

P16-L1/2: I guess the authors mean "between the period from the 1940s to the early 1990s and the period covering the 1990s to today"

P16-L5: I suggest the use of "sub-periods" instead of "periods"
P16-L5/6: Consider changing to: "…based on the predominance of different types of (re)sources available during the entire study period (Fig. 7)."
P16-L22: Change "exponential" to "considerable"
P16-L25: Consider starting the sentence with "The" instead of "These"
P16-L27: Instead of "…from when the registering of avalanches becomes more regular, a change linked…" consider using "…when avalanches were more regularly registered. This change is linked…"
P16-L31: Change "this data" to "then" and change "a winter" to "per winter"
P17-L9: Consider using "since" instead of "from"
P17-L10: Instead of "whatever", consider "regardless of"
P19-L8: Consider changing to "many small intensity events"
P21-L1: Consider "The increase of visits partly explains…"
P21-L9: Consider adding a reference: "…mid-twentieth century (Fig. 5B)…"
P22-L9: Change to "…from persona, experiences…"
P22-L11: Consider changing to: "this study was carried out in a" or "this study is associated to a"
P22-L12: Delete "already"
P22-L13: Instead of the "quality of the snow," I recommend to write "the characteristics of the snowpack," or "the structure of the snowpack,"
P22-L19: Consider deleting "global"
P22-L26: Delete "visible"
P23-L6: Consider changing to: "The absence or scarcity of snow is…"
P23-L21: Delete comma after "the analysis exploits"
P24-L26: Use a full stop instead of an exclamation mark
P25-L1: Change to "The reproduction of this type of…"

Authors' response: Thank you very much for all these suggestions and editing comments. They have all been incorporated in the revised manuscript. Note, however that due to text reworking and to further editing, some of the considered sentences may have disappeared or, at least, have been significantly modified.

**Author's response to Anonymous referee 2**

The paper presented by Giacona et al. deals with an interesting topic and even more addresses the approach to a region which has not been know so far for its avalanche activity. As such, the reconstruction presented is of high relevance, as these low elevation mountain ranges, as the Vosges, are likely to be the first and most severely affected by climate change and could thus serve as examples/illustrations of what one needs to expect at higher altitudes. Also, the number of sources collected by the authors is impressive and shows the great ability of the team to cross natural and human science approaches, which is still rare.
The main weaknesses of the paper reside in the way it was written, and this for several reasons. First of all, and despite the fact that the authors acknowledge a native speaker for the proof-reading, the text is written in a rather poor English, and many technical words have been translated simply from French (such that they either have a different or no more meaning in English). The language will need to be polished substantially in a new version.
The style of the paper also seems awkward to a natural scientist as it uses footnotes (and even in large numbers), which is all but normal in natural sciences (by contrast to human sciences). This needs to be clarified as well.
Thirdly, the manuscript is fairly descriptive in the introduction and not very clear either in the abstract. Overall, the text needs to become much more concise and focused, and also much clearer in view of terminology. What is a nonconventional source (for me, natural or written archives are conventional indeed, but not sufficiently used)?

Authors' response: We have deeply reworked all the formal aspects of the paper. Modifications made in the revised version of the paper we would gladly submit to your kind consideration include:

- A slight overall reorganization to better distinguish the content of the different sections. Also, the discussion section has been slightly expanded to put the results in perspective with regards to a larger context;
- A real effort for precising and defining the exact meaning of all the terms we are using, especially those related to the typology of avalanche events, and also those arising from the fields of history and social science (and therefore arguably less familiar to natural hazards (geo)scientists). To this end, we have included an additional table which contains the definition of such terms;
- Suppression of all footnotes, with inclusion of the relevant information within the text core;
- Yet, significant shortening of the text core;
- Correction of all typos, awkward sentences, etc., and further English smoothing by a professional English corrector.

However, we want to stress that the first author has her background in history, and that the main outcome of the paper, in addition to specific findings for the case study, is the contribution of this discipline to a better understanding of the evolution of natural hazards on the long range. This has two consequences for the paper:

- First, the data/result/discussion organization may remain a bit different from a pure geoscience paper. Specifically, analyzing the source amount, quality and evolution through time clearly belongs, for the historian, to the results section and not only to the discussion section. It is even one of the most important points of the work to illustrate that the two aspects cannot be truly distinguished;
- Second, the text style remains truly a bit more literary than in standard geoscience articles.

As stated before, we have polished the paper with regards to the first submission, which should contribute to make it easier and more convenient to read for the NHESS readership.

For instance, the additional table in the revised version of the paper should help making the bridge between historical / social science concepts and the field of natural hazards. Yet, we don't want to completely get rid of these intrinsic specificities of the work, which also contribute, in our opinion, to its value.

*Where do you address hazards, and where are you really addressing risks? This is used in a mixed way and needs clarification as well. What are risk historians? What is a geohistorical methodology/resources/approach? etc. (I could provide many more examples, but would like to suggest that the authors stick to the international literature when using definitions or terms.*

Authors' response: We adhere to the classical distinction between hazard and risk, where risk arises as the conjunction of the hazard (all possible avalanches, generally expressed as a probability distribution) with elements at risk such as humans, roads, etc. Yet, it must be stressed that our geo-chronology was built to be as close as possible of the reality of avalanches in the Vosges Mountains, which means that it includes both damageable and non-damageable events. As stated in the paper, old data are strongly biased towards damageable events whereas more recent ones include much more non-damageable events. As a consequence, our geo-chronology is neither a chronicle of avalanche hazard nor a chronicle of avalanche risk. Notably, it is clearly one of the main scopes of the discussion section to shade light on the different effects that combine to explain the actual shape of the event chronology, and to which extent it corresponds to a chronology of hazard and risk as function of time. We have made an additional effort through the whole text to clarify this point and specifically to distinguish the two concepts as much as possible.

For many words like *geo-history*, see the new table.

"Risk historians" was an awkward formulation. We meant historians working on risk due to natural hazards. This has been changed.

*The results will need to be presented in a much clearer, and more organized way. There is more in the data than you are showing so far.*

Authors' response: See before regarding the new organization.

By the way, we agree that we do not provide a detailed analysis of all the information the data convey. Specifically, we only perform a rapid analysis of its main features a function of time, geomorphology, events typology, sources, etc. The reason for this is that the main objective of the work is to illustrate how historical data such as ours, generally considered as too lacunar and uncertain to be exploited in the field of natural hazard, can be valued by a careful historical analysis. Of course, once this is done, and especially once the different effects that combine in the event geo-chronology have been discriminated, further exploitation of the data for, *e.g.*, climate inference, flow modelling, etc., becomes possible. Since the paper is already really long, we let this for further research, and we have added a more explicit remark in this direction in the conclusion/outlook section.

*In the same line of thoughts, please make sure that you put your data into a larger context, the discussion is very much focused at the case-study site so far and introduces many new results rather than seeing them in a broader context.*

Authors' response: As stated before, the paper has been slightly reorganized. The discussion section now includes a specific subsection with summarizes different elements that put our results in perspective with regards other contexts/studies. These include a few new elements and references and several elements already present in the first version of the paper but, we agree, in a too diffuse way. However, we want to stress that, as stated in introduction, studies as ours are very seldom, especially for avalanches, where knowledge

and published time series are mostly limited to last decades and on which very few historians have worked. This makes the value of our work higher, but also makes it more difficult the comparison of our results to other case-studies.

In my opinion the paper can become a very nice and relevant piece, and certainly will be suitable for NHESS, but more work is needed to reach this goal, and I would be happy to see a new version on that interesting topic sometimes soon.
Authors' response: We thank aging the referee for his encouragements and hope he will receive and enjoy the revised version of our paper.

---

## Author Response (AR2)

Cover letter for the revised and corrected version of our article: "*A 240 year History of Avalanche Risk in the Vosges Mountains from Nonconventional Sources*", submitted to Natural Hazards and Earth System Sciences

Dear Natural Hazards and Earth System Sciences editorial board, dear Dr. Sven Fuchs,

Thank you very much for for your kind comments about our work, and for noticing the typo in the figure caption of our draft. It has been corrected as suggested. We hope that that this will make the paper adequate for publication in Natural Hazards and Earth System Sciences, where we firmly believe that it should be of high interests for a large part of the readership.

With best regards,

Florie Giacona, for the authors.